# Reinforcement Learning via Self-Distillation

**Jonas Hübotter** [1]  **Frederike Lübeck** [* 1 2]  **Lejs Behric** [* 1]  **Anton Baumann** [* 1]  **Marco Bagatella** [1 2]  **Daniel Marta** [1]
**Ido Hakimi** [1]  **Idan Shenfeld** [3]  **Thomas Kleine Buening** [1]  **Carlos Guestrin** [4]  **Andreas Krause** [1]

## Abstract

Large language models are increasingly post-trained with reinforcement learning in verifiable domains such as code and math. Yet, current methods for reinforcement learning with verifiable rewards (RLVR) learn only from a scalar outcome reward per attempt, creating a severe credit-assignment bottleneck. Many verifiable environments actually provide rich textual feedback, such as runtime errors or judge evaluations, that explain *why* an attempt failed. We formalize this setting as reinforcement learning with rich feedback and introduce **Self-Distillation Policy Optimization** (**SDPO**), which converts tokenized feedback into a dense learning signal without any external teacher or explicit reward model. SDPO treats the current model conditioned on feedback as a self-teacher and distills its feedback-informed next-token predictions back into the policy. In this way, SDPO leverages the model's ability to retrospectively identify its own mistakes in-context. Across scientific reasoning, tool use, and competitive programming on LiveCodeBench v6, SDPO improves sample efficiency and final accuracy over strong RLVR baselines. Notably, SDPO also outperforms baselines in standard RLVR environments that only return scalar feedback by using successful rollouts as implicit feedback for failed attempts. Finally, applying SDPO to individual questions at test time accelerates discovery on difficult binary-reward tasks, achieving the same discovery probability as best-of-$k$ sampling or multi-turn conversations with $3\times$ fewer attempts.

Code: https://github.com/lasgroup/SDPO.

[1]ETH Zurich [2]Max Planck Institute for Intelligent Systems [3]MIT [4]Stanford University. Correspondence to: Jonas Hübotter <jonas.huebotter@inf.ethz.ch>.

*Proceedings of the 43rd International Conference on Machine Learning*, Seoul, South Korea. PMLR 306, 2026. Copyright 2026 by the author(s).

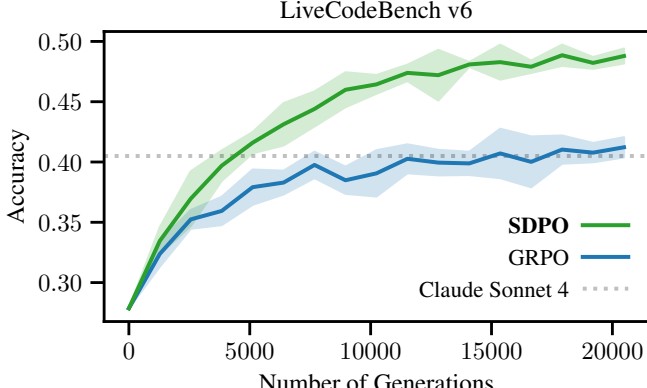

*Figure 1.* **SDPO substantially outperforms an improved version of Group Relative Policy Optimization (GRPO) on LCB v6 with Qwen3-8B.** Further, SDPO achieves GRPO's final accuracy in $4\times$ fewer generations. Claude Sonnet 4 is the strongest instruct model on the public LCBv6 leaderboard. Shaded regions show the standard deviation across 3 seeds.

## 1  Introduction

Progress in deep reinforcement learning has shown that iterating on experience—acting, receiving feedback, and updating a policy—can unlock capabilities that are difficult to obtain from static supervision alone (Mnih et al., 2015; Silver et al., 2016; 2017; Berner et al., 2019). The same theme now appears in large language models (LLMs): large-scale post-training with reinforcement learning (RL) has substantially improved performance on reasoning-heavy tasks, especially in settings with programmatic or otherwise verifiable evaluation (Jaech et al., 2024; Guo et al., 2025; Kimi et al., 2025; Olmo et al., 2025).

Nevertheless, the dominant RL recipe for LLM post-training remains bottlenecked by credit assignment. Most current approaches operate in the setting of reinforcement learning with verifiable rewards (RLVR): given a question $x$, the model samples an answer $y \sim \pi_\theta(\cdot \mid x)$ and receives a scalar reward $r \in \mathbb{R}$, often binary (e.g., unit-tests pass/fail in code generation). Modern policy gradient RLVR methods such as Group Relative Policy Optimization (GRPO; Shao et al., 2024) estimate advantages from these sparse outcome rewards. Furthermore, when all rollouts in a group receive the same (often zero) reward, GRPO advantages collapse

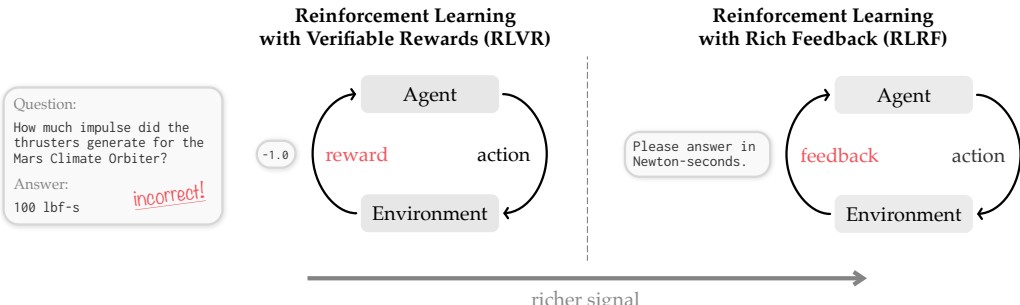

*Figure 2.* **Comparison of RLVR and RLRF settings.** In Reinforcement Learning with Verifiable Rewards (RLVR), the agent learns from a scalar reward, which often acts as an information bottleneck by masking the underlying environment state. In contrast, Reinforcement Learning with Rich Feedback (RLRF) utilizes tokenized feedback. This provides a significantly richer signal than a scalar reward, as the feedback can encapsulate a reward and observations of the state (e.g., runtime errors from a code environment or feedback from a judge).

to zero and learning stalls. To overcome this sparsity, one might prefer distillation from a strong teacher (Guo et al., 2025; Yang et al., 2025; Lu & Thinking Machines Lab, 2025; Guha et al., 2026), which provides dense, token-level supervision. However, strong teachers are often unavailable in online learning, where the goal is to raise the capability ceiling beyond existing models.

In this work, we argue that the key limitation is not RL per se, but the information bottleneck imposed by scalar outcome rewards. Many verifiable environments expose *rich tokenized feedback* beyond scalar rewards $r$, such as runtime errors, failing unit tests, or evaluations from an LLM judge. This feedback not only reveals *whether* a rollout was wrong, but also *what* went wrong. We formalize this more general setting as **Reinforcement Learning with Rich Feedback** (**RLRF**) and illustrate its difference to RLVR in Figure 2. Here, feedback can be any tokenized representation of any state reached by an agentic system. The central question becomes: how can we convert rich feedback into effective credit assignment without requiring external supervision from a strong teacher?

Our starting point is the observation that LLMs already possess a powerful mechanism for using feedback: in-context learning (Brown et al., 2020; Wei et al., 2022). When conditioned on feedback, the same model can often identify plausible mistakes and propose a corrected approach. A common example of such feedback is the summary of failed test cases on coding platforms like LeetCode (Figure 11 in the appendix). Many recent works leverage this capability to iteratively generate corrections (Chen et al., 2021a; Madaan et al., 2023; Shinn et al., 2023; Yao et al., 2024; Yuksekgonul et al., 2025; Lee et al., 2025). In contrast, we use the current policy as a "self-teacher" that, rather than sampling a new response, re-evaluates the *existing* rollout after receiving rich feedback. Including the feedback in-context transforms the model's next-token distribution, allowing the self-teacher to agree or disagree with the student's original choices at specific tokens. This yields dense, logit-level credit assignment.

| Method | Sampling | Signal | Feedback |
|---|---|---|---|
| **SFT / Distillation** | ✗ off-policy | ✓ rich | ✗ strong teacher |
| **On-Policy Distillation** | ✓ on-policy | ✓ rich | ✗ strong teacher |
| **RLVR (such as GRPO)** | ✓ on-policy | ✗ weak | ✓ environment |
| **RL via Self-Distillation** (ours) | ✓ on-policy | ✓ rich | ✓ environment |

*Table 1.* Comparison of self-distillation to alternative methods for post-training LLMs.

Crucially, this mechanism incurs no sampling overhead: we simply re-compute the log-probabilities of the original attempt under the self-teacher's feedback-augmented context.

Building on this idea, we introduce **Self-Distillation Policy Optimization** (**SDPO**), an on-policy algorithm that performs RL via self-distillation. SDPO samples rollouts from the current policy, obtains rich environment feedback, and then minimizes a logit-level distillation loss that matches the current policy's next-token distribution to that of the self-teacher. Conceptually, SDPO addresses the central limitation of applying distillation to online learning: the absence of a stronger external teacher. Instead of relying on a fixed teacher, SDPO leverages the model's ability to recognize its own mistakes in hindsight. By conditioning the current policy on the rich feedback it just received, we construct a self-teacher that provides the dense supervision of distillation while retaining the exploration benefits of on-policy RL. Table 1 summarizes how this positions SDPO relative to RLVR and distillation baselines. We include a summary of related work in Section 6.

We show that SDPO is a policy gradient algorithm whose advantages are estimated using the self-teacher. This enables the implementation of SDPO with minor changes to standard RLVR pipelines, simply by swapping out advantages.

**Summary of evaluation results:**

- **Learning without rich feedback** (§3): We evaluate standard RLVR environments that do not return any feedback beyond scalar rewards. Here, SDPO treats successful attempts sampled in the current batch as "feedback" for

failed attempts on the same question. We perform training runs on scientific reasoning and tool use, starting with Qwen3-8B and Olmo3-7B-Instruct. We find that SDPO outperforms a strong GRPO baseline that integrates recent improvements: 68.8% vs. 64.1% final accuracy on aggregate. SDPO achieves higher accuracy with up to 7× shorter generation lengths compared to GRPO, demonstrating that effective reasoning need not be verbose.

- **Learning with rich feedback** (§4): We evaluate competitive programming problems from LiveCodeBench v6 with LeetCode-style feedback. As shown in Figure 1, SDPO substantially improves over GRPO, reaching a higher final accuracy (48.8% vs. 41.2%) and achieving GRPO's final accuracy in 4× fewer generations. SDPO's gains grow with model scale, suggesting that the ability for self-teaching emerges as models become stronger in-context learners.
- **Discovering solutions to hard tasks at test-time** (§5): Finally, we demonstrate that SDPO can accelerate the discovery of solutions to difficult binary-reward questions. This contrasts with RLVR methods, which only begin learning once the first solution has been found. We leverage SDPO for **Test-Time Self-Distillation**, a form of test-time training where the model specializes to an individual test question. We consider very difficult LiveCodeBench questions, for which the base model's pass@64 is below 0.03, and show that SDPO accelerates the discovery of solutions by 3×.

## 2 Self-Distillation Policy Optimization

We propose an algorithm that uses the in-context learning ability of the current policy for assigning credit. Our key object is the *self-teacher*, $\pi_\theta(\cdot \mid x, f)$, which refers to the current policy (the "student") prompted with the question $x$ and the rich feedback $f$. Next to the student's original attempt $y$, $f$ may incorporate two key kinds of feedback: any environment output (such as runtime errors from a code environment) and a sample solution if $x$ was already solved with another attempt in the rollout group. As discussed before, the self-teacher $\pi_\theta(\cdot \mid x, f)$ should have a higher accuracy than the student $\pi_\theta(\cdot \mid x)$ since it sees additional information in-context. This leads us to observe:

**We can use the same policy in two different roles: As the student for the initial attempt and as the teacher to determine the value of actions in hindsight.**

We introduce **Self-Distillation Policy Optimization** (**SDPO**) which repeatedly distills the self-teacher into the student. Given a question $x$, we first sample rollouts from the student $\pi_\theta$ and obtain environment feedback. We use the KL-divergence, $\mathrm{KL}(p\|q) = \sum_i p(i) \log p(i)/q(i)$, as a distance measure for next-token distributions of student and teacher, and optimize a standard logit distillation loss:

$$\mathcal{L}(\theta) := \sum_t \mathrm{KL}(\pi_\theta(\cdot|x, y_{<t})\|\mathrm{sg}(\pi_\theta(\cdot|x, f, y_{<t}))) \quad (1)$$

where the stopgrad operator, sg, blocks gradients from flowing through the teacher, and thus prevents it from regressing towards the student and ignoring $f$. The intuitive role of the teacher is to determine where and how the students' original attempt $y$ was wrong through retrospection based on the feedback $f$. Figure 3 shows an example of self-teaching with Qwen3-8B as student and self-teacher. We summarize SDPO in Algorithm 1 in the appendix.

We derive the SDPO gradient as follows (cf. Appendix E.1):

**Proposition 2.1.** *The gradient of $\mathcal{L}$ is:*

$$\nabla_\theta \mathcal{L}(\theta) = \mathbb{E}_{y \sim \pi_\theta(\cdot|x)} \left[ \sum_{t=1}^{|y|} \mathbb{E}_{\hat{y}_t \sim \pi_\theta(\cdot|x, y_{<t})} \left[ \right. \right.$$
$$\left. \left. \log \frac{\pi_\theta(\hat{y}_t \mid x, y_{<t})}{\pi_\theta(\hat{y}_t \mid x, f, y_{<t})} \cdot \nabla_\theta \log \pi_\theta(\hat{y}_t \mid x, y_{<t}) \right] \right]. \quad (2)$$

### 2.1 Comparison to RLVR

Note that the SDPO gradient is a (negated) logit-level policy gradient where the advantages are estimated using the self-teacher (cf. Appendix E.4). We can therefore reuse standard RLVR implementations and simply swap out the advantages. Let $y_i$ be the $i$-th rollout from a rollout group of size $G$ for question $x$, then comparing GRPO and SDPO we have:

$$A_{i,t}^{\mathrm{GRPO}} := r_i - \mathrm{mean}\{r_i\}_{i=1}^G \text{ (constant in } t),$$
$$A_{i,t}^{\mathrm{SDPO}}(\hat{y}_{i,t}) = \log \frac{\pi_\theta(\hat{y}_{i,t} \mid x, f_i, y_{i,<t})}{\pi_\theta(\hat{y}_{i,t} \mid x, y_{i,<t})}.$$

The GRPO advantages are zero on any non-generated token and constant within a rollout $y_i$.[1] In contrast, the SDPO advantages are zero only for tokens where student and teacher perfectly agree. The SDPO advantage is positive for tokens which are more likely under the teacher while being negative for tokens which are less likely under the teacher. Thus, SDPO can be seen as a direct extension of standard RLVR methods in two ways:

1. from 1-bit feedback to *allowing arbitrary sequences of tokens as feedback*, and
2. leveraging this rich feedback to *estimate dense logit-level advantages*.

This tight connection to RLVR methods also enables a straightforward extension of the SDPO gradient from Equation (2) to off-policy data via PPO-style clipped importance sampling (Schulman et al., 2017), see Appendix E.4. We further show in Appendix C.1 that SDPO only incurs a minor

---

[1]We use unnormalized GRPO advantages (Liu et al., 2025b).

**SDPO: Self-Distillation Policy Optimization**

1. Question $x$
```
Write a python function that returns
all numbers from 1 to n. Answer
briefly.
```

2. Answer $y \sim \pi_\theta(\cdot \mid x)$
````
```python
def numbers_up_to_n(n):
    return list(range(1, n + 1))
```
````

3. Feedback $f$
```
Don't include n.
```

4. Credit assignment by **self-teacher $\pi_\theta(y \mid x, f)$**

*Figure 3.* Example of self-teaching with Qwen3-8B. The answer is generated by the model before seeing the feedback. Then, we re-evaluate the log-probs of the original attempt with the *self-teacher* after seeing the feedback. We show the per-token $\log\left(\mathbb{P}(\text{self-teacher})/\mathbb{P}(\text{student})\right)$, with red indicating negative values (self-teacher disagrees) and white indicating values around zero. Notably, in this example, Qwen3-8B identifies the error through retrospection without an explicit solution. Further, the activation is sparse, identifying where mistakes happen and adjusting to the students' response distribution.

compute overhead compared to GRPO and discuss stability improvements in Appendix C.2.

## 3 Learning without Environment Feedback

We first evaluate SDPO in standard RLVR environments, where feedback is limited to scalar rewards. Instead of using the scalar reward, SDPO treats successful attempts sampled in the current batch as "feedback" for failed attempts on the same question. By comparing the student's attempt with a correct solution, the self-teacher can identify where the student was wrong and provide dense credit assignment.

**Experimental setting**  We evaluate tasks on which the model has not been explicitly fine-tuned:

- **Science Q&A** (Chemistry, Physics, Biology, Materials science): Undergraduate-level scientific reasoning using reasoning subsets from SciKnowEval (Feng et al., 2024a).
- **Tool use**: Mapping an API specification and user request to the correct tool call (ToolAlpaca; Tang et al., 2023).

We perform a train-test split to test in-domain generalization. We use Qwen3-8B (Yang et al., 2025) and Olmo3-7B-Instruct (Olmo et al., 2025) as initial checkpoints and report avg@16 relative to wall-clock training time, excluding initialization & validation.

**Baselines.**  We compare SDPO to an improved variant of **GRPO** (Shao et al., 2024), which incorporates several recent modifications (cf. Appendix A). We additionally report the special case of **on-policy GRPO** (matching the hyperparameters of vanilla SDPO). For both baselines, we perform a hyperparameter sweep and report results for the models that achieve the highest validation performance across all target tasks. Hyperparameters and training details are provided in Appendix I. We use the `verl` library (Sheng et al., 2025) for fast multi-GPU training.

**Results**  Table 2 summarizes our results. We find that SDPO outperforms GRPO across almost all runs, often

| | Chemistry | | Physics | | Biology | | Materials | | Tool use | |
|---|---|---|---|---|---|---|---|---|---|---|
| | 1h | 5h | 1h | 5h | 1h | 5h | 1h | 5h | 1h | 5h |
| **Qwen3-8B** | 41.2 | | 59.2 | | 30.8 | | 58.9 | | 57.5 | |
| + GRPO | 65.9 | 74.5 | 63.8 | 72.7 | 35.1 | **59.9** | 74.3 | 77.1 | 64.9 | 67.7 |
| + GRPO (on-policy) | 63.3 | 63.4 | 63.6 | 63.6 | 49.8 | 49.8 | 73.9 | 74.1 | 60.2 | 65.7 |
| + **SDPO** (on-policy) | **73.2** | **80.9** | **66.6** | **75.6** | **50.6** | 56.8 | 72.1 | **78.4** | **68.0** | **68.5** |
| | | | | | | | | | | |
| **Olmo3-7B-Instruct** | 22.8 | | 37.7 | | 16.2 | | 36.7 | | 39.3 | |
| + GRPO | 39.7 | 56.7 | 55.3 | 63.3 | 35.6 | **55.8** | 70.9 | 75.0 | 56.4 | **65.0** |
| + GRPO (on-policy) | 51.4 | 57.5 | **62.7** | 62.7 | **49.8** | 49.8 | 73.3 | 73.5 | 56.8 | 60.6 |
| + **SDPO** (on-policy) | **68.0** | **80.0** | 59.9 | **66.1** | 48.0 | 52.8 | **73.7** | **79.1** | **60.8** | 62.1 |

*Table 2.* **Comparison of SDPO and GRPO on reasoning-related benchmarks.** We report the highest achieved avg@16 within 1 hour and 5 hours of wall-clock training time, respectively. Both SDPO and on-policy GRPO perform one gradient step per generation batch, while GRPO performs 4 off-policy mini batch steps. We select optimal hyperparameters for SDPO and baselines based on 5h accuracy. Each run is performed on a node with 4 NVIDIA GH200 GPUs. Together with initialization and validation, each run takes approximately 6 hours.

leading to substantial improvements. SDPO learns notably faster than GRPO, performing close to 5 hours of GRPO training after only 1 hour of training with SDPO in several cases. SDPO achieves a particularly substantial improvement over GRPO on the Chemistry task, as is displayed in Figure 4 (left). With Olmo3-7B-Instruct, *SDPO achieves the 5h GRPO accuracy in 50 minutes of wall-clock training time*, a $6\times$ speedup. Moreover, SDPO's 5h accuracy is more than 10%-points higher than that of GRPO.

We remark that our results with SDPO use strictly on-policy training (i.e., one gradient step per generation batch). Given the known efficiency gains of performing multiple gradient steps per generation batch, studying SDPO with off-policy updates is an exciting direction for future research.

**Self-distillation learns to reason concisely.**  We consistently observe that SDPO produces substantially shorter generations than GRPO while achieving higher accuracy. SDPO's responses are more than $3\times$ shorter on average across tasks (cf. Table 8 in Appendix H). On Chemistry with Olmo3-7B-Instruct, SDPO even achieves a $7\times$ reduction in

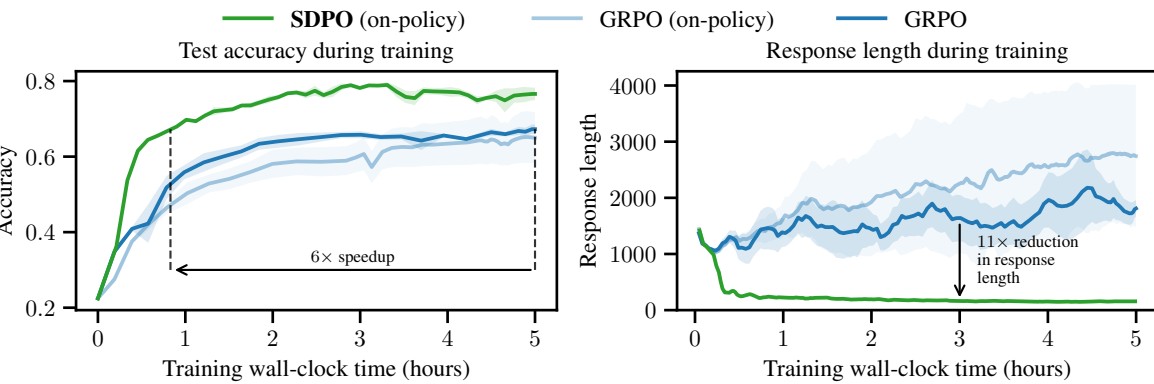

*Figure 4.* Training progression of Olmo3-7B-Instruct on Chemistry. We report the average accuracy across 16 samples per question and a rolling average of response lengths over 5 steps. We report GRPO with the optimal hyperparameters for this model and task.

---

...Alternatively... Closer to D? No... Wait I'm going in circles... Wait, perhaps the correct answer is B... $10^{1.85} \approx 69.3$... Ah, this works... Wait I think I messed up... Hmm... $10^{1.85} \approx 69.3$...
Thus, the correct answer is likely B: 1.85.
<answer>
B
</answer>

...At pH 7.4, all functional groups are neutral... maintaining a balance between hydrophobic and hydrophilic character... [The] overall polarity... keeps logD from being very high... or very low... [typically falling] in the 2.0-3.0 range, with 2.61 (C) being a reasonable estimate...
<answer>
C
</answer>

*(a) GRPO (5,549 tokens)*  *(b) SDPO (764 tokens)*

*Figure 5.* Example responses from GRPO and SDPO after 50 training steps to: "What is the correct octanol/water distribution coefficient logD under the circumstance of pH 7.4 for the molecule O=C1O[C@@H](COc2ccon2)CN1c1ccc(C2=CCOCC2)c(F)c1?" The answer options are A: 1.32, B: 1.85, C: 2.61, D: 3.76. The correct answer is **C**. GRPO's answer contains 5× "Hmm.", 9× "No.", and 25× "Wait". Further, GRPO's answer repeats calculations such as "$10^{1.85} \approx 69.3$", which appears four times, and the model even explicitly generates "Wait I'm going in circles". SDPO's answer avoids any circular reasoning and is more than 7× shorter. The base model is Qwen3-8B.

---

response length relative to GRPO while maintaining higher accuracy (Figure 4 (right)). While recent progress in RLVR has demonstrated that scaling response length is a powerful driver of emergent reasoning capabilities (Jaech et al., 2024; Guo et al., 2025; Muennighoff et al., 2025), our results suggest that effective reasoning need not always be verbose. We find that SDPO improves the *efficiency* of reasoning.

Qualitatively, we observe that the longer responses from GRPO often stem from "superficial" reasoning rather than necessary analytical steps. GRPO frequently generates filler phrases like "Hmm" and "Wait" or enters circular logical loops that repeat previous steps verbatim. Figure 5 displays a representative example of this phenomenon. Remarkably, SDPO's generations remain concise and avoid these superficial patterns. This may be explained by SDPO's dense credit assignment, which assigns a specific advantage to each next-token prediction, leading to sparse advantages (cf. Figure 21 in Appendix J). By improving the efficiency of reasoning, SDPO reduces inference generation time and demonstrates that reasoning performance can be improved by refining *how* the model reasons, not just how *long* it reasons.

## 4  Learning with Environment Feedback

We next evaluate SDPO on coding tasks. Coding is a canonical example of an RL environment that provides rich feedback, such as runtime errors and failed unit tests. Learning to solve these coding problems requires strong credit assignment since the student must identify its precise mistakes to avoid repeating them in the future. LiveCodeBench (LCB; Jain et al., 2025) provides a set of contest-style coding problems, ranging from simple to competition-level. We restrict our evaluation to the most recent LCBv6 subset of LCB, which contains 131 questions released between February and May 2025. We consider a setting with public and private unit tests, common for code contests and coding platforms like LeetCode, where the public tests are used for evaluation during training and the private tests are used for validation (Chen et al., 2022; Le et al., 2022; El-Kishky et al., 2025; Samadi et al., 2025).[2]

We use the Qwen3 (Yang et al., 2025) model family for our experiments, with Qwen3-8B as default unless otherwise specified. We report the average accuracy over 4 rollouts and use the same GRPO baseline as in Section 3.

---

[2]We select public tests as a 50% random subset of private tests.

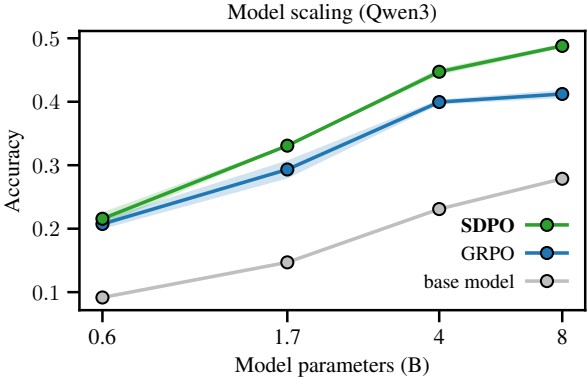

*Figure 6.* **SDPO improves with model size.** We compare the final LCBv6 validation accuracy of SDPO and GRPO at train step 80, across model sizes from Qwen3. The ability of SDPO's teacher to perform accurate retrospection appears to be an emergent phenomenon with scale. We include an additional scaling study with Qwen2.5-Instruct in the appendix (cf. Figure 17), supporting this finding. Error bars indicate the standard error across 3 seeds.

**Results.** Figure 1 compares the learning curves of SDPO and GRPO on LCBv6. We find that SDPO achieves a substantially higher final accuracy (48.8%) than GRPO (41.2%) while also outperforming the strongest instruct models on the public LCBv6 leaderboard:[3] Claude Sonnet 4 (40.5%) and Claude Opus 4 (39.7%). Furthermore, SDPO reaches the final accuracy of GRPO in $4\times$ fewer generations. We include an extended comparison to other RLVR baselines that perform similarly to GRPO in Table 9 in the appendix.

### 4.1 Self-distillation benefits from stronger models

A central question for our work is whether SDPO is sensitive to the in-context learning ability of the base model. Intuitively, we expect that SDPO benefits from a strong in-context learner, since this enables the teacher to perform more accurate retrospection.

To answer this question, we perform a scaling study with different model sizes from the Qwen3 (Yang et al., 2025) family. As shown by extensive prior work, the ability to learn in-context increases with model size (e.g., Brown et al., 2020). As depicted in Figure 6, SDPO significantly outperforms GRPO on larger models while only slightly improving over GRPO on smaller models. In a scaling study with the weaker Qwen2.5 model family, we observe the same trend, with SDPO underperforming GRPO on the weakest model (Qwen2.5-1.5B), as seen in Figure 17 in Appendix H.

### 4.2 Self-distillation performs dense credit assignment

Whereas GRPO assigns a constant advantage to each generated token, SDPO assigns an individual advantage to *each*

---

[3]On the public leaderboard, the LCBv6 subset can be obtained by selecting February to May 2025.

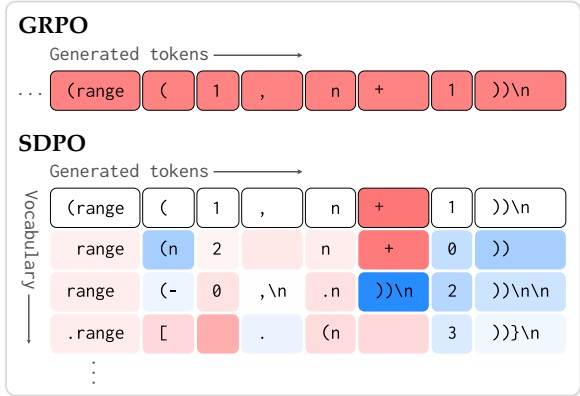

*Figure 7.* Dense credit assignment in SDPO in the example from Figure 3. Shown in blue are tokens which become more likely under the self-teacher. The self-teacher identifies how the returned range has to be modified so that it does not contain n.

*possible next token* along the generated sequence based on the agreement of student and teacher. At each position $t$ in the generated sequence $y$, there are $|\mathcal{V}|$ possible next tokens where $\mathcal{V}$ is the vocabulary. In distillation, this level is typically called the *logit-level* since it corresponds to the logits of the model. In practice, we approximate the full next-token distribution by the top-$K$ tokens, and as such, SDPO assigns $|y| \cdot K$ unique advantages per sequence. This is illustrated in Figure 7 and allows SDPO to perform dense credit assignment.

A natural question is whether the performance gains of SDPO are due to leveraging rich feedback in RLRF or due to the dense credit assignment of SDPO. To answer this question, we ablate three configurations:

- **Logit-level SDPO:** credit assignment over the 100 most likely tokens (under the student) at each position.
- **Token-level SDPO:** credit assignment over the most likely token at each position.
- **Sequence-level SDPO:** We compute SDPO advantages for all generated tokens and average them to produce a single scalar advantage per sequence (as in GRPO). This does not perform denser credit assignment than GRPO but still leverages the rich feedback $f$.

As shown in Figure 8 (left), the dense credit assignment of logit-level SDPO leads to significant performance gains over token-level SDPO and sequence-level SDPO. Nevertheless, even sequence-level SDPO outperforms GRPO, indicating that leveraging rich feedback in RLRF can lead to gains over RLVR methods even without dense credit assignment.

### 4.3 The self-teacher improves during training

Contrary to standard distillation, the self-teacher in SDPO is not frozen, but updated throughout training. This is a critical component of SDPO, since it enables the teacher to improve

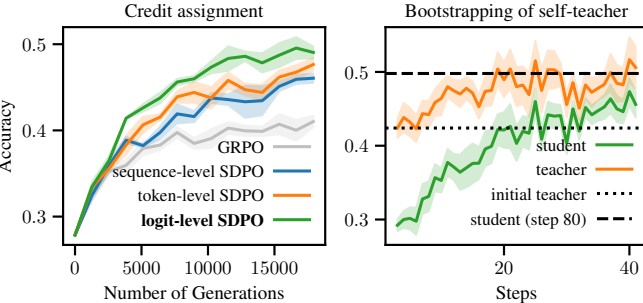

*Figure 8.* **Left: Rich feedback in RLRF and dense credit assignment of SDPO are complementary.** We compare logit-level, token-level, and sequence-level SDPO advantages to GRPO. While denser credit assignment in SDPO is beneficial (logit-level > token-level > sequence-level), even sequence-level SDPO significantly outperforms GRPO due to leveraging the rich feedback. Error bars indicate the standard error across 3 seeds. **Right: The self-teacher improves during training.** We display the generative accuracy of the self-teacher compared to student on the current training batch (with a rolling average over 5 steps). The final student score is taken at step 80. Notably, the performance of the student significantly surpasses the initial teacher's accuracy. Error bars indicate the standard deviation across 3 seeds.

over time, which means that the student can learn from a stronger target. To investigate whether the self-teacher improves during training, we plot the average accuracy when *generating* using the self-teacher in Figure 8 (right). We find that the self-teacher improves significantly during training. Most notably, the student's accuracy surpasses the initial teacher's accuracy in later stages of training. This demonstrates that SDPO enables bootstrapping of a weak model to a strong model, without the initial self-teacher's performance limiting the final student.

We further show in Appendix D that SDPO avoids catastrophic forgetting, and we analyze which feedback is most informative for the self-teacher.

## 5 Test-Time Self-Distillation

In Sections 3 and 4, we have demonstrated that SDPO can substantially improve over RLVR methods when performing "train-time RL" for reasoning tasks. We now turn to a test-time setting where the model is given only a single hard (binary-reward) question $x$ and must discover a solution as quickly as possible:

**Definition 5.1** (Discovery time). The discovery time is the number of trials needed until a solution is found (i.e., the smallest $k$ with the $k$-th attempt $y_k$ receiving reward 1).

Based on this notion, we can define a measure of the efficacy of discovery:

$$\begin{aligned} \text{discovery@}k &:= \mathbb{P}(\text{discovery time} \leq k) \\ &= \mathbb{P}(r(y_1 \mid x) = 1 \text{ or } \dots \text{ or } r(y_k \mid x) = 1), \end{aligned} \tag{3}$$

where the probability is over any randomness in the algorithm producing $y_k$ and the rewards. Thus, the discovery@$k$ metric quantifies the probability of discovering the solution within $k$ steps.[4] While prior work has studied discovery with continuous rewards (e.g., Novikov et al., 2025; Yuksekgonul et al., 2026), discovery with language models in sparse or binary-reward settings does not allow "hill-climbing" a continuous reward and has remained less well understood.

The most naive approach to discovery in binary-reward tasks is to sample repeatedly i.i.d. from the base model, also known as **best-of-**$k$. The canonical pass@$k$ metric for best-of-$k$ sampling is exactly the probability of discovering at least one solution within $k$ independent samples from a fixed model, coinciding with discovery@$k$. The discovery@$k$ metric generalizes pass@$k$ to algorithms that sample attempts sequentially. A common sequential approach re-prompts the base model with additional context from previous attempts (Madaan et al., 2023; Shinn et al., 2023). We refer to this as **multi-turn** sampling. Here, the model itself does not change, only its context evolves over time.

Performing RLVR on the question $x$ does not improve over best-of-$k$ sampling from the base model, since a binary reward provides no signal until the first solution has already been found.[5] An RLRF method like SDPO does not face the same limitation, as it receives rich feedback from the environment after each attempt. This rich feedback enables the model to repeatedly "correct" its mistakes as it encounters them and receives feedback, even before ever discovering a solution. In contrast to multi-turn sampling, SDPO repeatedly compresses context $c = (y_k, f_k)$ by distilling $\pi_\theta(\cdot \mid x, c)$ into a model $\pi_{\theta'}(\cdot \mid x)$ as we illustrate in Figure 9 in the appendix. This self-distillation enables SDPO to continually learn over long contexts, whereas the memory bottleneck of transformers inherently limits the context length of multi-turn sampling (Vaswani et al., 2017). In this section, we seek to answer the question: *Can repeatedly compressing context into model weights via self-distillation accelerate discovery for hard questions?*

**Experimental setting.** We consider a particularly challenging subset of questions from LCBv6 that are at Qwen3-8B's performance ceiling and require significant test-time sampling to find any solution. Concretely, we define two groups using Qwen3-8B's pass@$k$: *Hard tasks* with pass@$64 < 0.5$ and *very hard tasks* with pass@$64 < 0.03$. Among these, we retain questions for which *any* of best-of-$k$, multi-turn, or SDPO find at least one solution within

---

[4]Our proposed discovery@$k$ metric is a canonical metric in the study of runtime speedup (Dolan & Moré, 2002).

[5]For this reason, several works consider explicitly constructing curricula of solvable questions (e.g., Zhao et al., 2025; Huang et al., 2026; Diaz-Bone et al., 2025; Hübotter et al., 2025b), which self-distillation avoids. Other work found that RLVR yields limited improvement on hard questions (Yue et al., 2025).

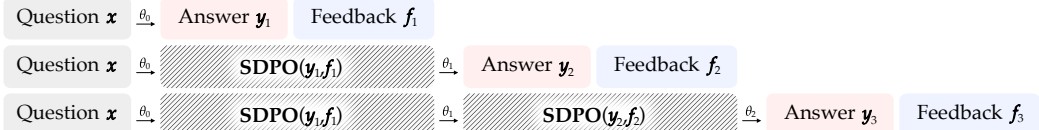

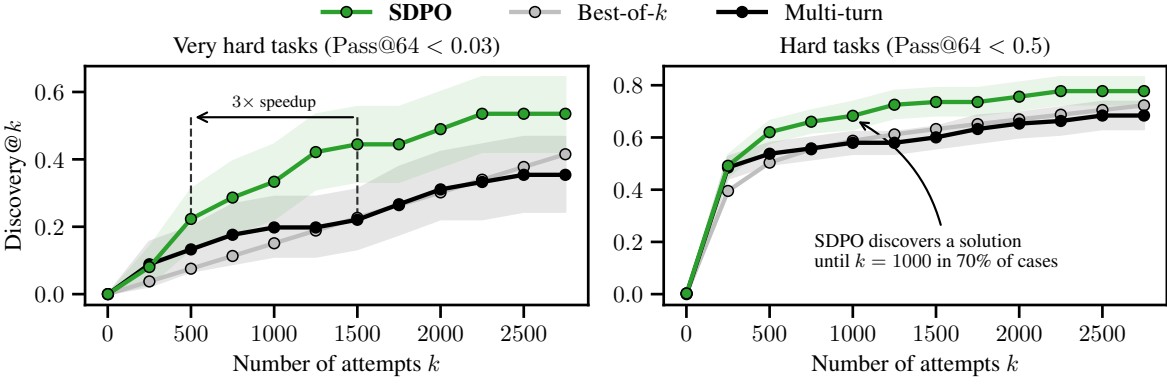

*Figure 9.* **Compressing context into model weights via self-distillation.** We illustrate the process of distilling the interaction history (context $c$) into the model parameters $\theta$. The model $\pi_\theta$ repeatedly attempts a fixed hard question $x$, generating an answer $y$ and receiving feedback $f$. Rather than appending this history to the context window, the model updates its weights $\theta_t \to \theta_{t+1}$ with SDPO (batch size 1) based on the feedback, effectively "fixing" mistakes by encoding $\pi_\theta(\cdot \mid x, c)$ directly into the policy $\pi_{\theta'}(\cdot \mid x)$.

*Figure 10.* **Self-distillation at test-time solves LiveCodeBench questions that neither the base model nor multi-turn conversations can solve. Left:** Very hard questions (9 total) from LCBv6 where the base model achieves pass@64 < 0.03, i.e., in less than 3% cases, sampling 64 responses yields any success. **Right:** Hard questions (19 total) from LCBv6 where the base model achieves pass@64 < 0.5. We report the discovery@$k$ metric, representing the probability of discovering at least one solution within $k$ total generations. Across both difficulty levels, SDPO achieves higher discovery@$k$ rates at almost all generation budgets, compared to the base model and a multi-turn conversation baseline that receives the feedback in-context. We report the mean and bootstrapped 90% confidence intervals of the mean across 5 random seeds per question.

512 steps across 5 seeds. Next to best-of-$k$ sampling, we compare SDPO to a multi-turn sampling baseline that sequentially receives the feedback in-context, using a first-in, first-out sliding window to remain within Qwen3-8B's 40k-token context limit. We evaluate SDPO with a batch size of 16 and ablate this in Figure 19 in Appendix H.

**Results.** Across both difficulty levels, SDPO achieves substantially higher discovery@$k$ rates at almost all generation budgets (Figure 10). On very hard tasks, SDPO achieves a discovery@2750 of 53.2%, significantly higher than the 41.5% and 35.6% achieved by best-of-k and multi-turn sampling, respectively. SDPO not only solves more questions overall but also does so with substantially fewer attempts. Notably, to reach a 22% discovery probability on very hard questions, SDPO requires 3× fewer generations than best-of-$k$ and multi-turn sampling. On hard tasks, SDPO reaches a 67% discovery probability with roughly 2.4× fewer generations than best-of-$k$ and multi-turn sampling. Beyond that, SDPO uniquely discovers a solution for Q3, which is neither solvable with multi-turn sampling nor with best-of-$k$ sampling within 2750 attempts. In contrast, SDPO first discovers a solution for Q3 after 321 attempts, which corresponds to 20 iteration steps of self-distillation based on feedback with a batch size of 16. We include detailed

per-question results in Table 10 in Appendix H.

Notably, the self-teacher's initial accuracy is < 1% for almost all questions, and even exactly 0% on 78% of them (Table 11 in Appendix H). This shows that a single turn of in-context feedback is insufficient to solve the problem. Despite this, the self-teacher's credit assignment is sufficiently effective for SDPO to iteratively refine the policy and eventually solve these questions.

## 6 Related Work

**Reinforcement Learning with LLMs.** Large-scale RL training has significantly advanced LLM reasoning capabilities (Guo et al., 2025; Kimi et al., 2025; Olmo et al., 2025), primarily via RLVR methods that estimate advantages from scalar outcome rewards (Zelikman et al., 2022; Shao et al., 2024). To overcome the information bottleneck of scalar signals, Process Reward Models (PRMs) have been proposed to provide step-level credit assignment (Lightman et al., 2023; Wang et al., 2024a). Unlike our RLRF setting, PRMs are typically trained on scalar rewards, either on value estimates for intermediate states or on outcome rewards (Cui et al., 2025). Our work shows that *each language model is implicitly a PRM* through retrospection if given rich feedback.

**Self-Distillation.** Distillation typically transfers capabilities from a strong external teacher to a student (Hinton et al., 2015; Agarwal et al., 2024). Self-distillation removes the dependency on external teachers by training the model to mimic its own predictions generated under additional context (Snell et al., 2022; Bai et al., 2022). While recent works apply self-distillation to environment feedback (Scheurer et al., 2023; Dou et al., 2024; Song et al., 2026), they use *off-policy* objectives, training the student on teacher generations. SDPO advances this by performing *on-policy* self-distillation: it trains the student specifically to avoid mistakes in its own generations using feedback-informed retrospection, which we show is critical for performance (cf. Appendix D.1). We discuss additional related work in Appendix G.

In a concurrent submission (included in the supplementary material), we study self-distillation for learning from demonstrations. Unlike our setting, where the agent interacts online with an environment to receive feedback, the other work focuses on offline learning from existing datasets. With complementary contributions, both works study self-distillation as a new learning paradigm for the post-training of LLMs.

## 7 Conclusion and Limitations

We introduced **Reinforcement Learning with Rich Feedback** (RLRF), a paradigm where environments provide tokenized feedback beyond scalar rewards, and argued that this removes a key information bottleneck of RLVR. We then proposed **Self-Distillation Policy Optimization** (SDPO), which uses the current policy as a feedback-conditioned *self-teacher* and distills its corrected log-probabilities into the student. This leverages the model's ability to learn from context for dense credit assignment. We further demonstrated that SDPO can be implemented as a minimal, drop-in modification to standard RLVR pipelines.

Empirically, SDPO demonstrates superior sample efficiency and wall-clock convergence compared to GRPO on reasoning tasks, even when training in standard RLVR environments without rich feedback. SDPO's gains grow with model scale, suggesting that the capacity for self-correction scales with the model's in-context learning capabilities. Moreover, we show that performing SDPO at test time on individual hard binary-reward tasks accelerates the discovery of solutions compared to strong baselines.

SDPO enables learning from rich feedback in a way that that is arguably closer to human cognition: utilizing precise outcomes rather than just binary rewards. By allowing the model to determine retrospectively how it should have acted, we demonstrate that language models can convert diverse tokenized feedback into effective self-supervision.

**Limitations.** Our findings show that SDPO's performance depends on a model's in-context learning ability, suggesting that SDPO is primarily applicable for RL-training stronger base models, while it can underperform GRPO on weaker models. Moreover, performance depends on the quality of the environment feedback. If the environment provides uninformative or misleading feedback, a model may not be able to learn from it through SDPO. Finally, SDPO adds a small computational overhead compared to GRPO for computing the log-probs of the retrospective model. While often negligible, this may be a larger overhead for smaller models with shorter generation lengths, where generation time is comparatively small.

## Impact Statement

This paper presents work whose goal is to advance the field of Machine Learning. There are many potential societal consequences of our work, none of which we feel must be specifically highlighted here.

## Acknowledgments

We would like to thank Akira Yoshiyama, Yassir Akram, Parnian Kassraie, Jonathan Thomm, Roman Vorushin, Afra Amini, Imanol Schlag, Yu Sun, and Moritz Hardt for helpful discussions. We thank Eduard Durech for helpful conversations regarding the scaling of RL fine-tuning and for his technical guidance on distributed infrastructure and long-context optimization. We are grateful to Ruixu Zhou from Tsinghua University & the Tencent Hunyuan Team for pointing out an error in the initially derived gradient estimator. Furthermore, we would like to thank Leander Diaz-Bone for supporting dataset generation.

This project was supported through the Swiss AI compute grant a156 and, in part, compute grant infra01. JH was supported by the Swiss National Science Foundation under NCCR Automation, grant agreement 51NF40 180545. FL and MB were supported by the ETH-MPI Center for Learning Systems. TKB and IH were supported by an ETH AI Center Postdoctoral Fellowship. DM was supported by the Knut and Alice Wallenberg Foundation.

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

# Appendices

## A  Baselines

We use an improved variant of GRPO which incorporates several recent modifications (Olmo et al., 2025; Khatri et al., 2026) such as asymmetric clipping (Yu et al., 2025), avoiding biased normalization (Liu et al., 2025b), and correcting for off-policy data when using efficient inference frameworks (Yao et al., 2025). We integrate these modifications into a GRPO implementation that represents a strong baseline, as detailed in Equation (12) in Appendix E.4. GRPO enables off-policy training through PPO's clipped importance weighting (Schulman et al., 2017).

## B  Figures

This section contains figures supporting the main text.

- Figure 11 shows an example for rich feedback in a code environment.
- Algorithm 1 shows the SDPO training loop.
- Table 3 shows the reprompt template for the self-teacher.

```
Runtime Error
ZeroDivisionError: division by zero
Line 73 in separateSquares (Solution.py)

Last Executed Input
[[26,30,2],[11,23,1]]
```

*Figure 11.* Example of feedback from our code environment, inspired by LeetCode. Listings 5, 6, and 7 in the appendix show examples of feedback in case of a wrong answer, a memory error, and an index error.

---

**Algorithm 1** SDPO training loop (self-distillation with rich feedback)

---

**Require:** Language model $\pi_\theta$; dataset of questions $x$; number of rollouts $G$ per question; environment that returns feedback $f$ for an attempt.

1: **repeat**
2:   Sample question $x$ from the dataset.
3:   Sample responses $\{y_i\}_{i=1}^{G} \sim \pi_\theta(\cdot \mid x)$.
4:   Evaluate each response to obtain feedback $\{f_i\}_{i=1}^{G}$.
5:   **Self-distillation (teacher = current policy):**
6:   Compute token log-probabilities $\log \pi_\theta(y_{i,t} \mid x, f_i, y_{i,<t})$
7:   Update $\theta$ by gradient descent on $\mathcal{L}_{\text{SDPO}}(\theta)$.
8: **until** converged

---

| **User:** | prompt |
|---|---|
| | Correct solution: successful_previous_rollout |
| | The following is feedback from your unsuccessful earlier attempt: environment_output |
| | Correctly solve the original question. |
| **Assistant:** | original_response |

*Table 3.* Template for self-teacher. prompt is replaced with the question. A sample solution previously generated by the student is substituted for successful_previous_rollout (if available for this question; otherwise the paragraph is skipped). environment_output is replaced with the environment output (see, e.g., Figure 11) from the models' original attempt (if it was not successful and there is no solution; otherwise the paragraph is skipped). If the models' original attempt was successful, this attempt is passed as the correct solution. original_response is replaced with the models' original attempt to re-evaluate its log-probabilities under the self-teacher.

# C  Extended Section 2

## C.1  Compute time & memory

The only computational overhead of SDPO compared to GRPO is the additional computation of log-probs from the self-teacher, which can be effectively parallelized and is substantially faster than sequential generation. Figure 12 compares the compute time of SDPO and GRPO. As expected, the compute overhead of SDPO is relatively small. Here, we use a micro batch size of 2;[6] compute time can be further reduced by using larger micro batch sizes.

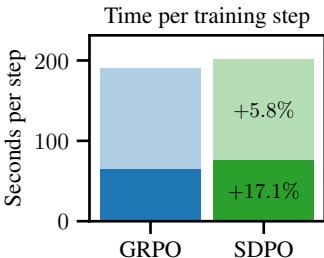

*Figure 12.* Time per step for SDPO vs GRPO (solid: without code environment, light: with code environment).

Naively computing the KL divergence between student and teacher requires holding full logits of both models in memory. To avoid this, we approximate the KL divergence in the SDPO loss by performing top-$K$ distillation (i.e., only computing the top-$K$ logits of the student and the corresponding logits of the teacher alongside a term capturing the tail probability; cf. Appendix E.3). With a reasonable choice of $K$ (e.g., $K = 100$), this avoids virtually any memory overhead while capturing most of the information.

## C.2  Stability improvements

We find that two practical modifications significantly enhance the training stability of SDPO. First, we employ a regularized self-teacher, implemented either via an exponential moving average (EMA) of the student parameters or by interpolating the current teacher with the initial teacher (cf. Appendix E.2). As detailed later, both strategies effectively stabilize learning. Second, we adopt the symmetric Jensen-Shannon divergence for the distillation loss; this formulation has similarly been shown to improve stability in on-policy distillation from external teachers (Agarwal et al., 2024).

---

[6]The micro batch size corresponds to # rollouts we train on at a time while accumulating gradients.

| | Task: | | Holdout tasks: | | | |
|---|---|---|---|---|---|---|
| | LCBv6 | IFEval | ArenaHard-v2 (hard prompt) | ArenaHard-v2 (creative writing) | MMLU-Pro | **Avg.** (holdout) |
| Base | 27.9 | 83.9 | 14.0 | 13.7 | 62.5 | 43.5 |
| SFT on self-teacher | 42.7 | 83.7 | 11.2 | 8.9 | 61.9 | 41.4 |
| GRPO | 41.2 | 82.2 | 12.0 | 10.8 | 62.3 | 41.8 |
| **SDPO** | 48.8 | 83.2 | 12.3 | 11.1 | 62.9 | 42.4 |

*Table 4.* **On-policy methods do not suffer from catastrophic forgetting.** We compare the accuracy of the final checkpoint on the training task LCBv6 and on holdout tasks IFEval, ArenaHard-v2, and MMLU-Pro. We compare to a baseline that trains directly on responses generated by the initial self-teacher with SFT. Overall, SDPO achieves the best performance–forgetting tradeoff. We include additional baseline results in Table 9 in the appendix.

# D    Extended Section 4

## D.1    On-policy self-distillation avoids catastrophic forgetting

Prior work has shown that a key benefit of on-policy algorithms, such as GRPO, is that models tend not to forget previously obtained capabilities (Shenfeld et al., 2026; Chen et al., 2025b; Lu & Thinking Machines Lab, 2025). This is practically desirable since it enables continual training pipelines where a model is trained sequentially on diverse tasks without the need to retrain from scratch. To evaluate forgetting, we test the final checkpoints of GRPO and SDPO on diverse holdout tasks: IFEval (Zhou et al., 2023), which tests the ability of a model to follow precise format instructions; ArenaHard-v2 (Li et al., 2025), which is an LLM-judged benchmark of real-world instruction-following prompts derived from LMArena (Chiang et al., 2024); and MMLU-Pro (Wang et al., 2024b), which tests broad multi-task knowledge and reasoning. As displayed in Table 4, SDPO learns the new task while mitigating degradation of initial capabilities, overall achieving a better performance–forgetting tradeoff than GRPO.

**Off-policy self-distillation baseline.**    As an additional baseline, we consider training the student via supervised fine-tuning (SFT) on successful generations from the self-teacher (Scheurer et al., 2023; Dou et al., 2024; Zhou et al., 2025).[7] This requires $2\times$ the generations of SDPO for the same number of steps, since we have to generate from both the student and the teacher. We report SFT on the successes of the self-teacher, which achieves a higher accuracy than also including initial successes from the student in the SFT data. As shown in Table 4, SFT on the self-teacher significantly underperforms SDPO on LCBv6, while leading to worse forgetting of prior capabilities. This mirrors prior findings on the instability of off-policy imitation (see, e.g., Agarwal et al., 2024).

## D.2    Can GRPO and SDPO be combined?

GRPO utilizes Monte Carlo advantages, which are unbiased with respect to the objective of maximizing expected reward $J(\theta) := \mathbb{E}_{y \sim \pi_\theta(\cdot|x)}[r(y \mid x)]$. In contrast, SDPO advantages are inherently biased with respect to $J(\theta)$ due to being computed from rich feedback and a self-teacher. This dichotomy parallels the fundamental distinction between Monte Carlo and bootstrapped advantages in RL: while the latter are biased, they typically yield lower variance (Sutton & Barto, 1998; Schulman et al., 2016). This motivates a hybrid approach that combines reward-derived GRPO advantages with feedback-derived SDPO advantages:

$$A_{i,t}^{\text{SDPO+GRPO}}(\hat{y}_{i,t}) := \lambda A_{i,t}^{\text{GRPO}}(\hat{y}_{i,t}) + (1-\lambda)A_{i,t}^{\text{SDPO}}(\hat{y}_{i,t}), \quad \lambda \in [0,1]. \tag{4}$$

As shown in Figure 13, SDPO+GRPO appears to be more robust to weaker models than SDPO. Intuitively, in a weaker model such as Qwen3-0.6B, the SDPO advantages are less reliable, and hence including the GRPO advantage helps to stabilize training. In contrast, we find that SDPO+GRPO slightly underperforms SDPO on stronger models such as Qwen3-8B. This suggests that the signal of GRPO, only informed by a scalar reward, can be actively harmful with a strong initial model.

---

[7]SFT on a teacher's predictions is a standard off-policy distillation approach (Kim & Rush, 2016).

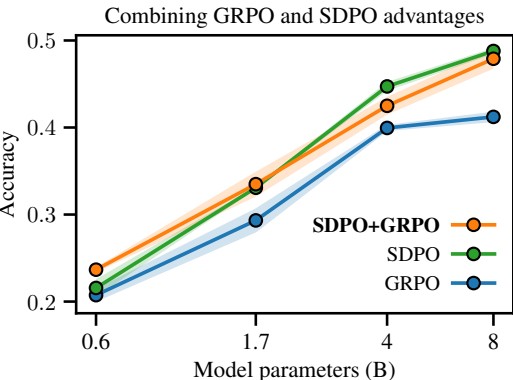

*Figure 13.* We compare the LCBv6 validation accuracy at step 80, across model sizes from Qwen3. SDPO+GRPO significantly outperforms SDPO on the weaker Qwen3-0.6B, while slightly underperforming SDPO on stronger models. We use $\lambda = 0.9$. Error bars indicate the standard error across 3 seeds.

| | Teacher before training | | Student trained with SDPO | |
|---|---|---|---|---|
| | $\uparrow$ Acc. (%) | $\downarrow$ Same output (%) | $\uparrow$ Acc. (%) | Avg. entropy |
| $f = $ output | $32.5 \pm 0.5$ | $13.7 \pm 0.6$ | $39.9 \pm 1.1$ | $0.40 \pm 0.0$ |
| $f = $ own solution | $\mathbf{42.4 \pm 1.0}$ | $12.1 \pm 0.7$ | $42.6 \pm 1.3$ | $0.41 \pm 0.0$ |
| $f = $ output + own solution | $\mathbf{42.5 \pm 1.2}$ | $\mathbf{10.1 \pm 0.2}$ | $\mathbf{48.3 \pm 1.4}$ | $0.38 \pm 0.0$ |
| $f = y$ + output + own solution | $39.3 \pm 0.8$ | $30.0 \pm 0.9$ | $44.5 \pm 1.3$ | *$0.23 \pm 0.0$* |

*Table 5.* **Performance of varying kinds of feedback.** We evaluate informativeness of feedback based on SDPO training (until step 70) as well as the direct impact on the self-teacher. "Same output" measures the percentage of cases where the teacher receives the same environment output as the student's initial attempt (i.e., not exploring alternative approaches). We observe that environment output and sample solutions are complementary and each provide informative feedback. Naively including only solutions or initial attempts $y$ significantly reduces diversity in the teacher and student. We remark that the sample solutions are generated by the student, enabling similar group-relative advantage estimation to GRPO. Error bars indicate standard deviation across 3 seeds.

### D.3  Which feedback is most informative?

To understand which type of rich feedback is most informative, we ablate the three types of feedback present in a verifiable environment like code generation: the sample solution (if a successful rollout is available in the current rollout group), the environment output (such as runtime errors), and the student's original attempt.

**Sample solutions.**  Including a sample solution from a failed attempt's rollout group (if available) closely mirrors the group-relative advantages of GRPO. We emphasize that these sample solutions are always generated by the student, as in GRPO, and do not require an expert model. They allow for disincentivizing unsuccessful approaches if the model is already able to solve the question. However, unlike GRPO where all tokens receive the same negative advantage, the self-teacher can identify specific mistakes and provide feedback on how to fix them.

**Environment output.**  The environment output describes the state of the environment after the student's attempt. This is complementary to sample solutions since it can provide useful signal even if the student has never solved the question before (a setting we explore extensively in Section 5). Leveraging environment output is a key differentiating factor between RLRF and RLVR settings.

**Student's original attempt.**  The student's original attempt $y$ does not have to be included in the reprompting template of the teacher. Indeed, we find that including it biases the teacher towards the student's attempt (cf. Table 5). This reduces the entropy of the student's distribution (particularly for initially uncertain tokens), thereby reducing exploration.

We summarize results in Table 5 where we evaluate the effect on SDPO training as well as the direct impact on the self-teacher. We find that environment output & sample solutions are complementary, each providing informative feedback. Generally, we observe that performance is not sensitive to syntactic variations of the reprompting template from Table 3.

| Teacher | Accuracy | Avg accuracy |
|---|---|---|
| $q_\theta$ | $36.1 \pm 1.6$ | $29.8 \pm 1.3$ |
| $q_{\theta_{\text{ref}}}$ | $48.8 \pm 0.7$ | $44.4 \pm 0.2$ |
| Trust-region | $\mathbf{50.6} \pm 0.9$ | $\mathbf{45.6} \pm 0.2$ |
| EMA | $49.3 \pm 0.3$ | $\mathbf{45.3} \pm 0.2$ |

*Table 6.* Best/average accuracy until step 90 of various methods for teacher regularization. Trust-region and EMA teachers use $\alpha = 0.01$. Training of the $q_\theta$ eventually diverges. Error ranges indicate standard errors across 3 seeds.

### D.4   Teacher regularization improves training stability

As described in Appendix C.2, SDPO uses a regularized teacher to stabilize training. As can be seen in Table 6, a non-regularized teacher significantly underperforms the regularized teachers. Furthermore, trust-region and EMA teachers outperform the teacher frozen at the initial teacher's parameters, showing that the teacher improves through parameter sharing with the student. Yet, SDPO performs well even with a frozen teacher.

# E Implementation of SDPO

The following pseudocode in Figure 14 outlines the implementation of SDPO:

```python
def compute_sdpo_loss(batch, teacher_context, loss_mask):
    """
    Computes probabilities of response y under the self-teacher
        and the per-logit SDPO loss.
    """
    # Compute model probabilities for response y
    logprobs_student = compute_log_prob(batch) # (T,V)
    probs_student = logprobs_student.exp() # (T,V)

    # Compute self-teacher probabilities for response y
    teacher_batch = reprompt(batch, teacher_context)
    logprobs_teacher = compute_log_prob(teacher_batch).detach() # (T,V)

    # Compute SDPO loss: per-token divergence
    per_token_loss = divergence(logprobs_student, logprobs_teacher) # (T,)
    return agg_loss(per_token_loss, loss_mask, loss_agg_mode="token-mean")
```

*Figure 14.* The pseudo-code of SDPO within a standard RL training pipeline. Omitted here is the filtering to top-$K$ logprobs for student and teacher (including a tail term) as described in Appendix E.3. Further, we omit here any importance sampling weights to correct for off-policy data. `reprompt` modifies the batch to incorporate teacher context (i.e., rich feedback). `divergence` implements any per-token divergence such as reverse-KL, forward-KL, or Jensen-Shannon.

In the following, we provide further details on:

- The gradient estimator used in our implementation (Appendix E.1)
- Teacher regularization (Appendix E.2)
- Approximating logit-distillation with the top-$K$ logits for saving GPU memory (Appendix E.3)
- Generalizing PPO-style policy gradient algorithms to logit-level advantages (Appendix E.4)

To disambiguate the notation of the self-teacher, we use $q_\theta(\cdot \mid x, f) := \pi_\theta(\cdot \mid \text{reprompt}(x, f))$ in the following. Here, `reprompt` denotes the reprompt template of the self-teacher.

## E.1 Gradient Estimators

In this seciton, we discuss two possible gradient estimators for the KL divergence between the current policy $\pi_\theta(y \mid x)$ and the teacher policy $q_\theta(y \mid x, f)$.

**Per-token estimator.** Deriving the gradient of the SDPO loss as defined in Equation (1):

$$\mathcal{L}_{\text{token}}(\theta) := \mathbb{E}_{y \sim \text{stopgrad}(\pi_\theta(\cdot|x))} \left[ \sum_{t=1}^{T} \text{KL}(\pi_\theta(\cdot \mid x, y_{<t}) \| \text{stopgrad}(\pi_\theta(\cdot \mid x, f, y_{<t}))) \right] \tag{5}$$

leads to the following estimator (see a detailed proof in Appendix F.1), which corresponds to the sum of gradients of the KL divergence at each token:

$$\nabla \mathcal{L}_{\text{token}}(\theta) = \mathbb{E}_{y \sim \pi_\theta(\cdot|x)} \left[ \sum_{t=1}^{T} \mathbb{E}_{\hat{y}_t \sim \pi_\theta(\cdot|x, y_{<t})} \left[ \nabla_\theta \log \pi_\theta(\hat{y}_t \mid x, y_{<t}) \cdot \log \frac{\pi_\theta(\hat{y}_t \mid x, y_{<t})}{\pi_\theta(\hat{y}_t \mid x, f, y_{<t})} \right] \right]. \tag{6}$$

This corresponds to the estimator presented in Proposition 2.1. This gradient estimator effectively assumes that the sampling distribution generating $y$ is fixed.

**Sequence-level estimator.** An alternative self-distillation objective minimizes the sequence-level KL divergence between student and self-teacher, i.e.,

$$
\begin{aligned}
\mathcal{L}_{\text{seq}}(\theta) := \text{KL}(\pi_\theta \| q_\theta) &= \mathbb{E}_{y \sim \pi_\theta(\cdot \mid x)} \left[ \log \frac{\pi_\theta(y \mid x)}{q_\theta(y \mid x, f)} \right] \\
&= \sum_{t=1}^{T} \mathbb{E}_{s_t \sim \Pi_\theta} \left[ \text{KL}(\pi_\theta(\cdot \mid s_t) \| q_\theta(\cdot \mid s_t, f)) \right],
\end{aligned}
\tag{7}
$$

where $s_t = (x, y_{<t})$ is the prefix ("state") at step $t$ and $\Pi_\theta$ denotes the prefix distribution under policy $\pi_\theta$. Estimating the gradient of this objective additionally takes into account how the choice of $y_t$ influences future states $y_{>t}$ (due to the additional dependence on $\Pi_\theta$).

Amini et al. (2025) show that the corresponding gradient estimator is given by

$$
\boldsymbol{\nabla}\mathcal{L}_{\text{seq}}(\theta) = \boldsymbol{\nabla}\mathcal{L}_{\text{token}}(\theta) + \mathbb{E}_{y \sim \pi_\theta(\cdot \mid x)} \left[ \sum_{t=1}^{T} \text{KL}(\pi_\theta(\cdot \mid s_t) \| q_\theta(\cdot \mid s_t, f)) \boldsymbol{\nabla}_\theta \log \Pi_\theta(s_t) \right].
\tag{8}
$$

The additional term of the sequence-level gradient captures how prefixes influence the self-distillation divergence of future tokens. We also experimented with this sequence-level gradient estimator but did not find measurable gains relative to its additional complexity.

### E.2 Regularized teacher

In contrast to standard distillation, the teacher in SDPO changes throughout training. This bootstrapping enables the teacher to improve, but it may also lead to training instability. To stabilize training, we seek to prevent the teacher $q$ from quickly diverging from the initial teacher $q_{\theta_{\text{ref}}}$. We can achieve this by placing an explicit trust-region constraint on $q$ (Schulman et al., 2015; Peng et al., 2019), that is:

$$
\sum_{t} \text{KL}(q(y_t \mid x, f, y_{<t}) \| q_{\theta_{\text{ref}}}(y_t \mid x, f, y_{<t})) \leq \epsilon, \quad \epsilon > 0.
\tag{9}
$$

This trust-region can be implemented in two ways:

1. **Explicit trust-region:** We can define the teacher as the policy closest to $q_\theta$ while satisfying the trust-region constraint. This teacher can be expressed as

$$
q(y_t \mid x, f, y_{<t}) \propto \exp\big((1 - \alpha) \log q_{\theta_{\text{ref}}}(y_t \mid x, f, y_{<t}) + \alpha \log q_\theta(y_t \mid x, f, y_{<t})\big),
\tag{10}
$$

with $\alpha \in (0, 1)$ the inverse Lagrange multiplier for the trust-region constraint. We include a full derivation in Appendix F.2. We can plug this explicitly constrained teacher directly into the SDPO objective.

2. **Exponential moving average (EMA):** Alternatively, we can stabilize the teacher's parameters directly; parameterizing $q_{\theta'}$ by $\theta'$ and updating as $\theta' \leftarrow (1 - \alpha)\theta' + \alpha\theta$ with $\alpha \in (0, 1)$.

Note that each implementation has a different practical advantage: The EMA teacher requires additional GPU memory for $\theta'$ yet does not introduce any runtime overhead. In contrast, the trust-region teacher requires an additional log-prob computation with $q_{\theta_{\text{ref}}}$ yet does not require additional GPU memory if $\theta_{\text{ref}}$ is used for explicit KL regularization.

### E.3 Approximate Logit Distillation

To save GPU memory, we perform distillation only on the top-$K$ tokens predicted by the student:

$$\mathcal{L}_{\text{SDPO}}(\theta) = \sum_{t=1}^{T} \text{KL}(\pi_\theta(\cdot \mid x, y_{<t}) \| \text{stopgrad}(q_\theta(\cdot \mid x, f, y_{<t})))$$

$$\approx \sum_{t=1}^{T} \sum_{\hat{y}_t \in \text{top}_K(\pi_\theta)} \pi_\theta(\hat{y}_t \mid x, y_{<t}) \cdot \log \frac{\pi_\theta(\hat{y}_t \mid x, y_{<t})}{\text{stopgrad}(q_\theta(\hat{y}_t \mid x, f, y_{<t}))} \tag{11}$$

$$+ \left(1 - \sum_{\hat{y}_t \in \text{top}_K(\pi_\theta)} \pi_\theta(\hat{y}_t \mid x, y_{<t})\right) \cdot \log \underbrace{\frac{1 - \sum_{\hat{y}_t \in \text{top}_K(\pi_\theta)} \pi_\theta(\hat{y}_t \mid x, y_{<t})}{\text{stopgrad}\left(1 - \sum_{\hat{y}_t \in \text{top}_K(\pi_\theta)} q_\theta(\hat{y}_t \mid x, f, y_{<t})\right)}}_{\text{tail}}$$

Here, the top-$K$ is with respect to student. Without top-$K$ distillation, we would have to keep two copies of logits in memory: one for teacher and student each. Top-$K$ distillation avoids virtually any memory overhead without impacting performance significantly, since most tokens of the vocabulary are not informative at a given time.

### E.4 Off-Policy Training: Generalization to Logit-Level Losses

PPO-style clipping (Schulman et al., 2017) with truncated importance sampling (Yao et al., 2025), clip-higher (Yu et al., 2025), fixed length normalization (Liu et al., 2025b):

$$\mathcal{L}_{\text{token}}(\theta) := -\frac{1}{\sum_{i=1}^{G} |y_i|} \sum_{i=1}^{G} \sum_{t=1}^{|y_i|} \min\left(w_{i,t}^{\text{TIS}}, \rho\right) \min\left(w_{i,t} A_{i,t}, \text{clip}(w_{i,t}, 1 - \varepsilon_{\text{low}}, 1 + \varepsilon_{\text{high}}) A_{i,t}\right), \tag{12}$$

with $w_{i,t} := \frac{\pi_\theta(y_{i,t} \mid x, y_{i,<t})}{\pi_{\theta_{\text{old}}}(y_{i,t} \mid x, y_{i,<t})}$, $w_{i,t}^{\text{TIS}} := \frac{\pi_{\theta_{\text{old}}}(y_{i,t} \mid x, y_{i,<t})}{\pi_{\theta_{\text{old}}}^{\text{rollout}}(y_{i,t} \mid x, y_{i,<t})}$, and $A_{i,t}$ denotes the per-token advantage.

We extend this to a logit-level loss:

$$\mathcal{L}_{\text{logit}}(\theta) := -\frac{1}{\sum_{i=1}^{G} |y_i|} \sum_{i=1}^{G} \sum_{t=1}^{|y_i|} \sum_{\hat{y}_{i,t}} \min\left(\pi_{\theta_{\text{old}}}(\hat{y}_{i,t} \mid x, y_{i,<t}), \rho \pi_{\theta_{\text{old}}}^{\text{rollout}}(\hat{y}_{i,t} \mid x, y_{i,<t})\right)$$
$$\min\left(w_{i,t}(\hat{y}_{i,t}) A_{i,t}(\hat{y}_{i,t}), \text{clip}(w_{i,t}(\hat{y}_{i,t}), 1 - \varepsilon_{\text{low}}, 1 + \varepsilon_{\text{high}}) A_{i,t}(\hat{y}_{i,t})\right), \tag{13}$$

where $\hat{y}_{i,t}$ sums over all possible tokens at position $t$ for rollout $i$ (or the $K$ most likely under $\pi_{\theta_{\text{old}}}$, cf. Appendix E.3). The TIS changes since we explicitly weight each logit by its probability under $\pi_{\theta_{\text{old}}}$ rather than relying on a Monte Carlo estimate of the expectation over next-token predictions. Here, $A_{i,t}(\hat{y}_{i,t})$ is a per-logit advantage.

In our experiments for SDPO, we apply the TIS term on a token-level rather than logit-level.

# F Theoretical Analysis

This section is organized as follows:

- Appendix F.1 derives the SDPO gradient from Theorem 2.1.

- Appendix F.2 derives the trust-region regularized teacher discussed in Appendix E.2.

To disambiguate the notation of the self-teacher, we use $q_\theta(\cdot \mid x, f) := \pi_\theta(\cdot \mid \mathrm{reprompt}(x, f))$ in the following. Here, `reprompt` denotes the reprompt template of the self-teacher.

## F.1 Proof of Proposition 2.1.

*Proof.* In the following, we derive the gradient of $\mathcal{L}_{\mathrm{SDPO}}$.

$$\nabla_\theta \mathcal{L}_{\mathrm{SDPO}}(\theta) = \nabla_\theta \sum_{t=1}^{T} \mathrm{KL}(\pi_\theta(\cdot \mid x, y_{<t}) \| \mathrm{stopgrad}(q_\theta(\cdot \mid x, f, y_{<t})))$$

$$= \nabla_\theta \sum_{t=1}^{T} \sum_{\hat{y}_t} \pi_\theta(\hat{y}_t \mid x, y_{<t}) \log \left( \frac{\pi_\theta(\hat{y}_t \mid x, y_{<t})}{\mathrm{stopgrad}(q_\theta(\hat{y}_t \mid x, f, y_{<t}))} \right)$$

Let $A_{t,k} := \log \left( \frac{\mathrm{stopgrad}(q_\theta(\hat{y}_t \mid x, f, y_{<t}))}{\pi_\theta(\hat{y}_t \mid x, y_{<t})} \right)$. Then,

$$= -\nabla_\theta \sum_{t=1}^{T} \sum_{\hat{y}_t} \pi_\theta(\hat{y}_t \mid x, y_{<t}) A_{t,k}$$

$$= -\sum_{t=1}^{T} \sum_{\hat{y}_t} \pi_\theta(\hat{y}_t \mid x, y_{<t}) \nabla_\theta A_{t,k} + A_{t,k} \nabla_\theta \pi_\theta(\hat{y}_t \mid x, y_{<t}).$$

We have that $\nabla_\theta A_{t,k} = -\nabla_\theta \log \pi_\theta(\hat{y}_t \mid x, y_{<t})$ is the negative score function. Using the score trick, $\pi_\theta(\hat{y}_t \mid x, y_{<t}) \nabla_\theta \log \pi_\theta(\hat{y}_t \mid x, y_{<t}) = \nabla_\theta \pi_\theta(\hat{y}_t \mid x, y_{<t})$. Hence, the first term simplifies to

$$-\sum_{t=1}^{T} \sum_{\hat{y}_t} \pi_\theta(\hat{y}_t \mid x, y_{<t}) \nabla_\theta A_{t,k} = \sum_{t=1}^{T} \sum_{\hat{y}_t} \nabla_\theta \pi_\theta(\hat{y}_t \mid x, y_{<t}) = \sum_{t=1}^{T} \nabla_\theta \underbrace{\sum_{\hat{y}_t} \pi_\theta(\hat{y}_t \mid x, y_{<t})}_{=1} = 0.$$

Thus, the gradient of $\mathcal{L}_{\mathrm{SDPO}}$ is

$$\nabla_\theta \mathcal{L}_{\mathrm{SDPO}} = -\sum_{t=1}^{T} \sum_{\hat{y}_t} A_{t,k} \nabla_\theta \pi_\theta(\hat{y}_t \mid x, y_{<t})$$

$$= -\sum_{t=1}^{T} \sum_{\hat{y}_t} \pi_\theta(\hat{y}_t \mid x, y_{<t}) \Big( A_{t,k} \nabla_\theta \log \pi_\theta(\hat{y}_t \mid x, y_{<t}) \Big)$$

$$= -\sum_{t=1}^{T} \mathbb{E}_{\hat{y}_t \sim \pi_\theta(\cdot \mid x, y_{<t})} \left[ A_{t,k} \nabla_\theta \log \pi_\theta(\hat{y}_t \mid x, y_{<t}) \right].$$

$\square$

Notably, the above implies that the gradient of $\mathcal{L}_{\mathrm{SDPO}}$ is equivalent to the gradient of the loss if $A_{t,k} = \mathrm{stopgrad} \left( \log \frac{q_\theta(y_t \mid x, f, y_{<t})}{\pi_\theta(y_t \mid x, y_{<t})} \right)$.

## F.2 Trust-region Teacher

To stabilize training, we seek to prevent the teacher $q$ from diverging from the initial teacher $q_{\theta_{\text{ref}}}$. We can achieve this by placing an explicit trust-region constraint on the teacher $q$ (Schulman et al., 2015; Peng et al., 2019), that is:

$$\sum_t \text{KL}(q(y_t \mid x, f, y_{<t}) \| q_{\theta_{\text{ref}}}(y_t \mid x, f, y_{<t})) \leq \epsilon, \quad \epsilon > 0. \tag{14}$$

In the following, we derive a teacher $q$ which satisfies the trust-region constraint while staying close to the target $q_\theta$. The following optimization problem characterizes such a $q$ (Peng et al., 2019):

$$\begin{aligned} \arg\max_{q \in \Delta} \quad & \sum_t \sum_{y_t} q(y_t \mid x, f, y_{<t}) \log \frac{q_\theta(y_t \mid x, f, y_{<t})}{q_{\theta_{\text{ref}}}(y_t \mid x, f, y_{<t})} \\ \text{s.t.} \quad & \sum_t \text{KL}(q(y_t \mid x, f, y_{<t}) \| q_{\theta_{\text{ref}}}(y_t \mid x, f, y_{<t})) \leq \epsilon, \end{aligned} \tag{15}$$

where $\Delta$ denotes the probability simplex. Intuitively, the solution is the $q$ satisfying the trust-region constraint, which is closest to $q_\theta$ (i.e., has minimal cross-entropy to $q_\theta$) while being farthest from $q_{\theta_{\text{ref}}}$ (i.e., has maximal cross-entropy to $q_{\theta_{\text{ref}}}$).

**Proposition F.1.** *The solution to Equation* (15) *can be expressed in closed form as*

$$q^*(y_t \mid x, f, y_{<t}) \propto \exp\big((1-\alpha) \log q_{\theta_{\text{ref}}}(y_t \mid x, f, y_{<t}) + \alpha \log q_\theta(y_t \mid x, f, y_{<t})\big). \tag{16}$$

*Proof.* To simplify notation, we omit the conditioning in the following. The Lagrangian (with $\lambda \geq 0$ for the KL constraint and $\nu$ for normalization) is

$$\mathcal{L}(q, \lambda, \nu) = \sum_t \sum_{y_t} q(y_t) \log \frac{q_\theta(y_t)}{q_{\theta_{\text{ref}}}(y_t)} - \lambda \Big( \sum_{y_t} q(y_t) \log \frac{q(y_t)}{q_{\theta_{\text{ref}}}(y_t)} - \epsilon \Big) + \nu \Big( \sum_{y_t} q(y_t) - 1 \Big).$$

Stationarity gives, for all $y_t$,

$$0 = \frac{\partial \mathcal{L}}{\partial q(y_t)} = \log \frac{q_\theta(y_t)}{q_{\theta_{\text{ref}}}(y_t)} - \lambda \Big( \log \frac{q(y_t)}{q_{\theta_{\text{ref}}}(y_t)} + 1 \Big) + \nu.$$

Let $\alpha := 1/\lambda$. Then, the solution to Equation (15) can be characterized in closed form as

$$\begin{aligned} q^*(y_t) &\propto q_{\theta_{\text{ref}}}(y_t) \exp\Big( \alpha \log \tfrac{q_\theta(y_t)}{q_{\theta_{\text{ref}}}(y_t)} \Big) \\ &\propto \exp\big((1-\alpha) \log q_{\theta_{\text{ref}}}(y_t) + \alpha \log q_\theta(y_t)\big). \end{aligned}$$

$\square$

(Chen et al., 2025c) perform a similar derivation, but use reference $\pi_{\theta_{\text{ref}}}$, which we observe to underperform compared to the reference $q_{\theta_{\text{ref}}}$.

# G Additional Related Work

In this section, we provide additional context and literature beyond Section 6 from the main body.

**Extended discussion on RLVR.** Standard RLVR methods, such as STaR or GRPO (Zelikman et al., 2022; Shao et al., 2024), operate similarly to the classical REINFORCE algorithm (Williams, 1992). While some approaches utilize separate value networks to reduce variance (Schulman et al., 2017), these incur substantial memory costs during training. Conceptually, our work is related to "bootstrapping your own latent" (BYOL; Grill et al., 2020) and "expert iteration" (Anthony et al., 2017) where a student is bootstrapped by repeatedly imitating an improved version of itself (called the "expert"). Canonically, the expert combines the student with test-time search, such as tree search (Anthony et al., 2017) or majority voting (Zuo et al., 2025). In contrast, SDPO leverages the student's ability to learn from rich feedback provided in-context, which is related to "augmented views" in BYOL.

**Extended discussion on Distillation.** Traditional distillation transfers capabilities by mimicking either the output distribution or intermediate representations of a fixed teacher (Romero et al., 2015; Kim & Rush, 2016; Sanh et al., 2019; Xie et al., 2020). Recent on-policy approaches have focused on mitigating train-test distribution shifts (Gu et al., 2024; Yang et al., 2025; Lu & Thinking Machines Lab, 2025), a challenge closely related to online imitation learning (Ross et al., 2011).

**Extended discussion on Self-Distillation.** Self-distillation has proven effective for diverse tasks such as compressing factual information (Eyuboglu et al., 2026; Kujanpää et al., 2025; Cao et al., 2025a) or prompt-based behaviors (Choi et al., 2022) into model weights. Regarding feedback-based learning, prior off-policy methods (Mitra & Ulukus, 2025) differ fundamentally from SDPO. Specifically, off-policy self-distillation optimizes the student to mimic the teacher's success, whereas SDPO's on-policy formulation optimizes the student to correct its own specific failures. We also note concurrent work by (Chen et al., 2025c), who apply on-policy self-distillation in grid world settings with scalar rewards. Their method uses a reflection stage to diagnose mistakes, showing that self-distillation can offer improved credit assignment over learned value networks even in simpler, non-linguistic domains.

## G.1 Learning from Rich Feedback and through Retrospection

Beyond scalar outcome rewards, recent works have leveraged rich execution or verbal feedback to guide generation (Gehring et al., 2025; Feng et al., 2024b; Yuksekgonul et al., 2025). A primary line of research focuses on translating verbal feedback into reward functions for RL. This is often achieved by mapping feedback to discrete token-level rewards using an external frozen model (Wang et al., 2026), or by employing strong external LLMs to explicitly construct state-wise reward functions (Goyal et al., 2019; Xie et al., 2024; Urcelay et al., 2026).

Alternatively, feedback can be utilized without explicit reward modeling. Several approaches focus on in-context improvement without integrating the process into the RL optimization loop (Chen et al., 2021a; Madaan et al., 2023; Shinn et al., 2023; Yao et al., 2024; Yuksekgonul et al., 2025; Lee et al., 2025). Others manually curate preference datasets by pairing responses before and after feedback to train with direct preference optimization (Stephan et al., 2024; Lee et al., 2024), though this requires additional generation and lacks the direct credit assignment of SDPO. Various recent works bootstrap thinking traces from known answers, using these answers as rich feedback (Zhou et al., 2026; Hatamizadeh et al., 2026; Zhang et al., 2025).

A central object in several recent works is a feedback-conditioned policy $\pi_\theta(y \mid x, f)$, which learns answers $y$ that lead to feedback $f$ (Liu et al., 2023; Zhang et al., 2023; Luo et al., 2025), typically through supervised objectives. The idea behind these approaches is to deploy a policy conditioned on desirable (i.e., positive) feedback for deployment. This approach is conceptually related to goal-conditioned RL (Schaul et al., 2015; Liu et al., 2025a), where one can learn from negative examples through goal relabeling (Andrychowicz et al., 2017). Feedback-conditioned policies view feedback as a goal, whereas RLRF views feedback as a state that can be used to determine whether the goal $x$ is achieved. Unlike SDPO, these methods do not use feedback for credit assignment in negative trajectories, but rather as a data transformation for goal relabeling.

## G.2 Further related work

**Value networks and Monte Carlo advantage estimation.** Several prior approaches aim to improve credit assignment but face the same information bottleneck as GRPO. Classical RL frequently trains value networks which provide token-level advantages, but themselves are learned from scalar rewards (Schulman et al., 2016; 2017). Furthermore, value networks

incur significant computational and memory overhead and are therefore typically not used to train LLMs. Other recent work estimates token-level advantages by performing additional generations starting from various positions in the original attempt (Kazemnejad et al., 2025; Zheng et al., 2025b). While this can learn with fewer gradient steps than GRPO it still uses only scalar rewards as signal and requires costly additional generations.

**Dense credit assignment with a reward model.** Several recent works have explored assigning dense (per-token) rewards given access to an external reward model, leveraging internal structure of the reward model (Chan et al., 2024; Cao et al., 2025b).

**Partial observability.** From the perspective of classical RL, many verifiable domains for LLMs are naturally *partially observable*: executing a proposed solution induces a latent environment state (e.g., failing tests or states of an agentic system) that is revealed only through rich feedback. This aligns with the formalism of partially observable Markov decision processes (POMDPs), where agents must act under incomplete observations of state (Kaelbling et al., 1998; Sutton & Barto, 1998). By contrast, RLVR and RLHF pipelines typically discard this observation channel and learn only from terminal scalar rewards or pairwise preferences.

**Relation to test-time training.** Our setting from Section 5 can be seen as a special case of test-time training where the model itself is updated at test-time using self-distillation. Updating the model at test-time is known as test-time training (Sun et al., 2020; 2025; Hardt & Sun, 2024; Hübotter et al., 2025a;b; Akyürek et al., 2025; Behrouz et al., 2025; Tandon et al., 2025; Hübotter et al., 2026). Unlike prior work, self-distillation uses the in-context learning ability of the current model to attribute credit after receiving feedback. This can be seen as simulating long-context reasoning with periodic compression of context into the model weights.

### G.3 SDPO as Maximum Entropy RL

The SDPO objective resembles the objective in maximum entropy RL (e.g., Levine, 2018; Haarnoja et al., 2018) with a particular choice of reward function.

**Maximum Entropy RL** Consider optimizing

$$\arg\max_{\theta} \; \mathbb{E}_{y \sim \pi_{\theta}(\cdot \mid x)} \left[ \sum_t r(y_t \mid x, y_{<t}) \right] + \lambda \mathrm{H}[\pi_{\theta}(\cdot \mid x)], \quad \lambda > 0 \tag{17}$$

where $\pi_{\theta}(y \mid x) = \prod_{t=1}^{T} \pi_{\theta}(y_t \mid x, y_{<t})$ and $\mathrm{H}[\pi_{\theta}(\cdot \mid x)] = \mathbb{E}_{y \sim \pi_{\theta}(\cdot \mid x)}[-\log \pi_{\theta}(y \mid x)]$ is the entropy of the policy. Here, $r(y_t \mid x, y_{<t})$ is an arbitrary reward function, possibly "dense" (i.e., per-token). Equation (17) is known as maximum entropy RL. It is known that this objective is equivalent to solving a variational inference problem which discuss next.

To this end, we define a Bernoulli random variable $\mathcal{C}$ which is 1 if the attempt $y$ is correct and 0 otherwise. We then define its distribution as $p(\mathcal{C} = 1 \mid x, y) \propto \exp(\frac{1}{\lambda} \sum_t r(y_t \mid x, y_{<t}))$. Further assuming w.l.o.g. that the "prior" over responses is uniform, we can express the posterior conditioned on the event of correctness as

$$\pi^{\star}(y \mid x) := p(y \mid x, \mathcal{C} = 1) \propto p(\mathcal{C} = 1 \mid x, y) \propto \exp\left( \frac{1}{\lambda} \sum_t r(y_t \mid x, y_{<t}) \right). \tag{18}$$

Then, Equation (17) is equivalent to minimizing the KL divergence with respect to $\pi^{\star}$:

$$\arg\min_{\theta} \; \sum_t \mathrm{KL}(\pi_{\theta}(y_t \mid x, y_{<t}) \| \pi^{\star}(y_t \mid x, y_{<t})). \tag{19}$$

**SDPO optimizes an implicit reward defined by the teacher** Note that Equation (19) is equivalent to the SDPO objective (Equation (1)) with implicit reward $r(y_t \mid x, y_{<t}) = \log q(y_t \mid x, f, y_{<t})$ and $\lambda = 1$. In this sense, SDPO can be seen as a maximum entropy RL algorithm with dense rewards constructed implicitly through the retrospective model.

This also points to a connection of SDPO to inverse RL (Ng et al., 2000; Ziebart et al., 2008; Rafailov et al., 2023), where the goal is to recover an unknown reward function. In SDPO, the student learns an implicit reward function defined by the retrospective model.

# H    Additional Results & Ablations

This section is organized as follows:

- Appendix H.1 contains results and ablations for Section 3.
- Appendix H.2 contains results and ablations for Section 4.
- Appendix H.3 contains results and ablations for Section 5.

## H.1    Learning without rich environment feedback

- Table 7 reports results when optimal hyperparameters are selected for each model/task combination.
- Table 8 compares average response lengths of SDPO and GRPO.

| | Chemistry | | Physics | | Biology | | Materials | | Tool use | |
|---|---|---|---|---|---|---|---|---|---|---|
| | 1h | 5h | 1h | 5h | 1h | 5h | 1h | 5h | 1h | 5h |
| **Qwen3-8B** | 35.6 | | 59.2 | | 27.9 | | 58.9 | | 57.5 | |
| + GRPO | 54.2 | 69.6 | 62.9 | 74.5 | 34.3 | 51.8 | **74.3** | 77.1 | **61.7** | 68.1 |
| + GRPO (on-policy) | 54.2 | 69.6 | 62.9 | 74.8 | 30.3 | 49.4 | 73.3 | 75.8 | **61.7** | 68.1 |
| + **SDPO** (on-policy) | **59.9** | **70.1** | **70.6** | **80.6** | **53.1** | **53.1** | 72.1 | **78.3** | 56.4 | **68.5** |
| **Olmo3-7B-Instruct** | 18.8 | | 37.7 | | 18.1 | | 36.7 | | 39.3 | |
| + GRPO | 42.7 | 54.3 | 55.3 | 63.3 | 54.2 | **63.8** | 73.8 | 78.1 | 56.4 | **65.0** |
| + GRPO (on-policy) | 48.8 | 54.3 | **62.7** | 62.7 | 54.2 | **63.8** | 67.9 | 74.4 | 56.0 | 61.3 |
| + **SDPO** (on-policy) | **59.2** | **76.8** | 60.3 | **71.4** | **56.1** | 58.3 | **75.3** | **79.2** | **57.3** | 62.5 |

*Table 7*. **Comparison of SDPO and GRPO on reasoning-related benchmarks.** We report the highest achieved avg@16 within 1 hour and 5 hours of wall-clock training time, respectively. Both SDPO and on-policy GRPO perform one gradient step per generation batch, while GRPO performs 4 off-policy mini batch steps. We select optimal hyperparameters for SDPO and baselines based on 5h accuracy. We perform this selection independently for each model and dataset. Each run is performed on a node with 4 NVIDIA GH200 GPUs. Together with initialization and validation, each run takes approximately 6 hours. *As opposed to Table 2 which selects globally optimal hyperparameters per method, this table selects optimal hyperparameters individually for each model/task combination based on 5h accuracy.* The hyperparameter grid is described in Section I.2.1.

| Model | GRPO | SDPO | Reduction of SDPO |
|---|---|---|---|
| **Qwen3-8B** | 820.8 | 255.8 | 3.2× |
| **Olmo3-7B-Instruct** | 1095.4 | 343.9 | 3.2× |

*Table 8*. Average response lengths of SDPO and GRPO (averaged across tasks from Section 3). Both algorithms are evaluated in the on-policy setting.

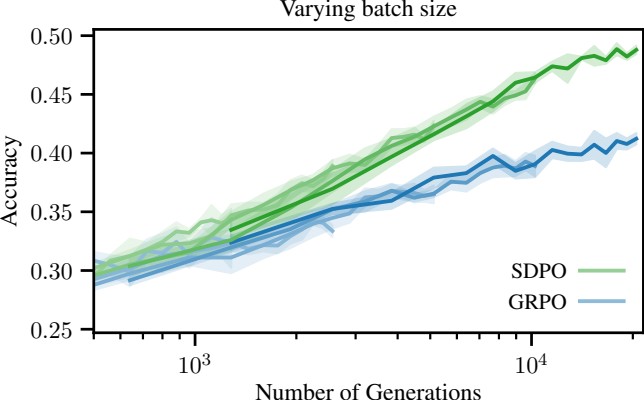

*Figure 16.* Accuracy (pass@1) for varying train batch sizes (4, 8, 16, 32) and number of rollouts (4, 8) for training SDPO and GRPO with Qwen3-8B (Yang et al., 2025) on LCBv6, ± stderr across 3 seeds. Different shades of the same color correspond to different runs.

## H.2 Learning with rich environment feedback

### H.2.1 ADDITIONAL RESULTS

Figure 15 shows the average accuracy of SDPO and GRPO stratified by question difficulty. LCB differentiates between easy, medium, and hard questions. As displayed, SDPO significantly improves over GRPO in solving medium and hard questions, highlighting the importance of rich feedback for challenging tasks. Note that this categorization of questions is different from the one in Section 5.

In Figure 16, we compare different train batch sizes and number of rollouts for training GRPO and SDPO on LCBv6.

Complementing the results shown in Figure 6, we show additional results using Qwen2.5-Instruct (Qwen et al., 2024) in Figure 17.

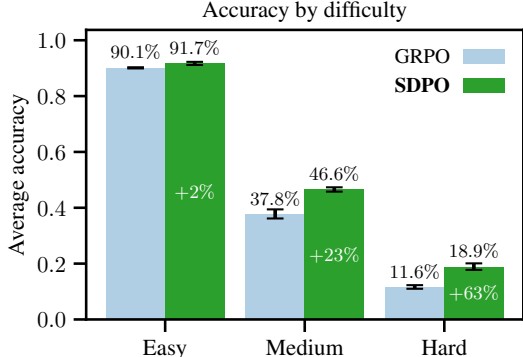

*Figure 15.* Average accuracy during training until step 80, stratified by difficulty. Error bars show standard deviation across 3 seeds.

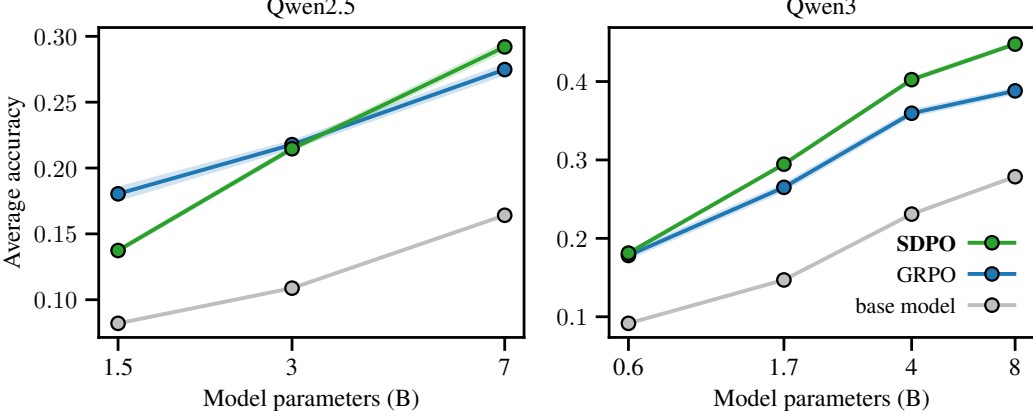

*Figure 17.* Average validation accuracy by model size, ± std across 3 seeds. With Qwen2.5-Instruct (Qwen et al., 2024) and Qwen3 (Yang et al., 2025) on LCBv6. Until step 65 for Qwen2.5 and until step 80 for Qwen3.

### H.2.2  TRAINING STABILITY

Figure 18 shows diverse metrics logged during training, including the loss, entropy, average gradient norm, and average response length.

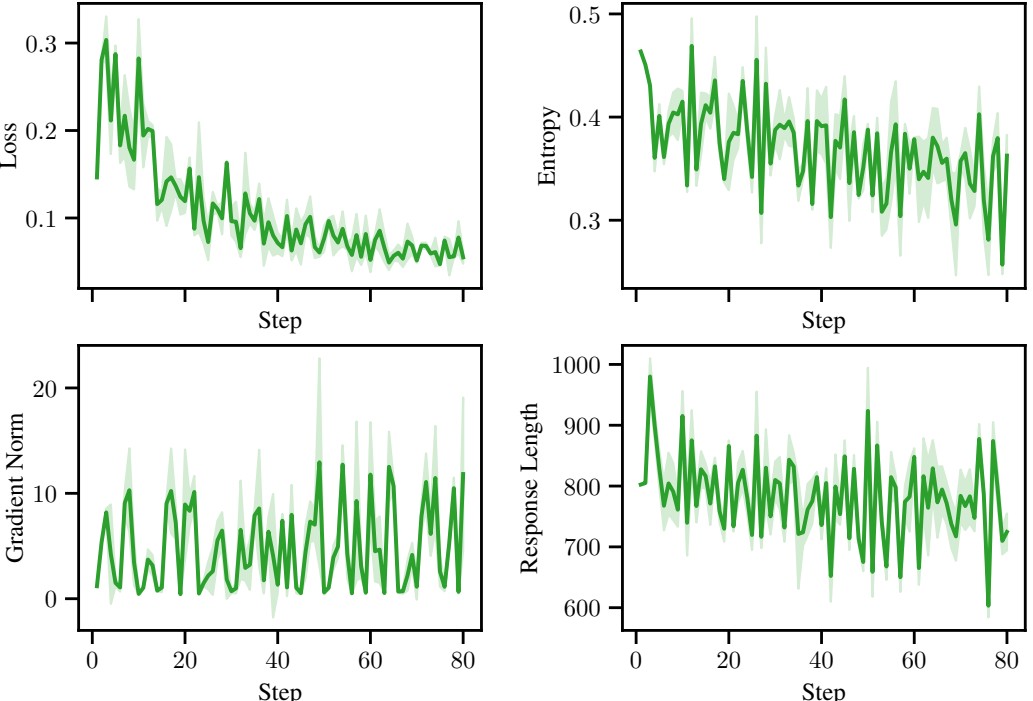

*Figure 18.* Loss, entropy, avg. gradient norm and avg. response length during training of SDPO on LCBv6 (Section 4

.

### H.2.3  BASELINES

Table 9 compares the performance on LCBv6 of various baselines, including two variants of GRPO, GSPO, and CISPO to SDPO.

|  | Accuracy | Avg accuracy |
| --- | --- | --- |
| GRPO | $41.2 \pm 0.8$ | $38.2 \pm 0.0$ |
| + only high-entropy tokens (Wang et al., 2025) | $37.8 \pm 2.2$ | $35.9 \pm 0.1$ |
| GSPO (Zheng et al., 2025a) | $40.1 \pm 2.3$ | $37.7 \pm 0.1$ |
| CISPO (Chen et al., 2025a) | $41.2 \pm 1.8$ | $37.8 \pm 0.1$ |
| **SDPO** | $\mathbf{48.8 \pm 0.6}$ | $\mathbf{43.8 \pm 0.0}$ |

*Table 9.* Performance on LCBv6 at/until training step 80 with std over 3 seeds. We compare to GSPO (Zheng et al., 2025a) and CISPO (Chen et al., 2025a). With Qwen3-8B.

### H.3  Test-time self-distillation

**Baselines**  For best-of-$k$ sampling under the base model, we report the standard pass@$k$ estimate (Chen et al., 2021b) from 2944 independent rollouts. As multi-turn sampling, we sequentially reprompt the model in-context using the concatenated feedback from previous attempts. To remain within Qwen3-8B's 40k-token context limit, we employ a first-in, first-out sliding window, discarding the earliest feedback once the maximum prompt length (32k tokens) is reached. We ablate the multi-turn reprompting strategy in Figure 19, and find that retaining only past feedback while forgetting earlier attempts significantly outperforms the baseline that additionally retains past attempts. We evaluate SDPO with a batch size of 16. We

| Question | SDPO | Best-of-$k$ | Multi-turn | Speedup Best-of-$k \rightarrow$ SDPO |
|---|---|---|---|---|
| 1 | 104 | 98 | **59** | 0.9× |
| 3* | **1987** | ≥ 2750 | ≥ 2750 | 1.4× |
| 10* | **938** | ≥ 2750 | 1706 | 2.9× |
| 43 | 111 | **109** | 111 | 1.0× |
| 46* | 1852 | 1466 | **1315** | 0.8× |
| 59 | 172 | 123 | **76** | 0.7× |
| 69 | 280 | **134** | **134** | 0.5× |
| 74* | 1948 | **1466** | 2405 | 0.8× |
| 86 | **85** | 421 | 335 | 5.0× |
| 91* | **1360** | ≥ 2750 | 2384 | 2.0× |
| 92* | **1575** | ≥ 2750 | 2203 | 1.8× |
| 95* | 1948 | **1466** | 1794 | 0.8× |
| 100 | **277** | 294 | 1596 | 1.1× |
| 103* | 2246 | ≥ 2750 | **2210** | 1.2× |
| 111 | 85 | 95 | **39** | 1.1× |
| 120 | **24** | 327 | 70 | 13.6× |
| 125* | 1795 | **1466** | 2320 | 0.8× |
| 127 | **28** | 368 | 61 | 13.1× |
| 129 | 168 | 173 | **104** | 1.0× |
| Hard tasks | **894** | 1145 | 1141 | 1.3× |
| Very hard tasks | **1739** | 2180 | 2121 | 1.2× |

*Table 10.* Mean number of generations until first success per question for SDPO, best-of-$k$ sampling, and the multi-turn sampling. For the mean calculation, values are truncated at the maximum budget of 2750 generations. Very hard tasks (pass@64 < 0.03) are marked with an asterisk (*). Averaged over all questions, SDPO achieves successes faster than the baselines, reaching a speedup of up to 13.6× on individual questions compared to best-of-$k$ sampling.

ablate this choice in Figure 19 and find that overall performance differences are marginal.

**Results for individual questions** We show the discovery@$k$ curves for all hard questions in Figure 20, and report the mean number of generations until the first discovery in Table 10. Further, Table 11 shows the per-question accuracy of the self-teacher at the initial training step of SDPO.

**Ablations** In Figure 19, we ablate the choice of batch size for SDPO and the in-context reprompting strategy for multi-turn sampling. We find that overall performance differences are marginal, yet smaller batch sizes are beneficial for improvements at low generation budgets, while larger batch sizes result in more stable updates that still learn to solve questions at later stages into the run.

**Multi-turn context length** The context window length for multi-turn sampling is reached after 837 (±466) steps for hard questions and after 1007 (±349) steps for very hard questions, offering a possible explanation for its diminishing gains at high generation budgets.

**Further Details** In the selection of hard questions, we have discarded one malformed question (Q9) where the coding environment did not correctly validate the solution due to rounding inaccuracies, which led to failures even with correct logic.

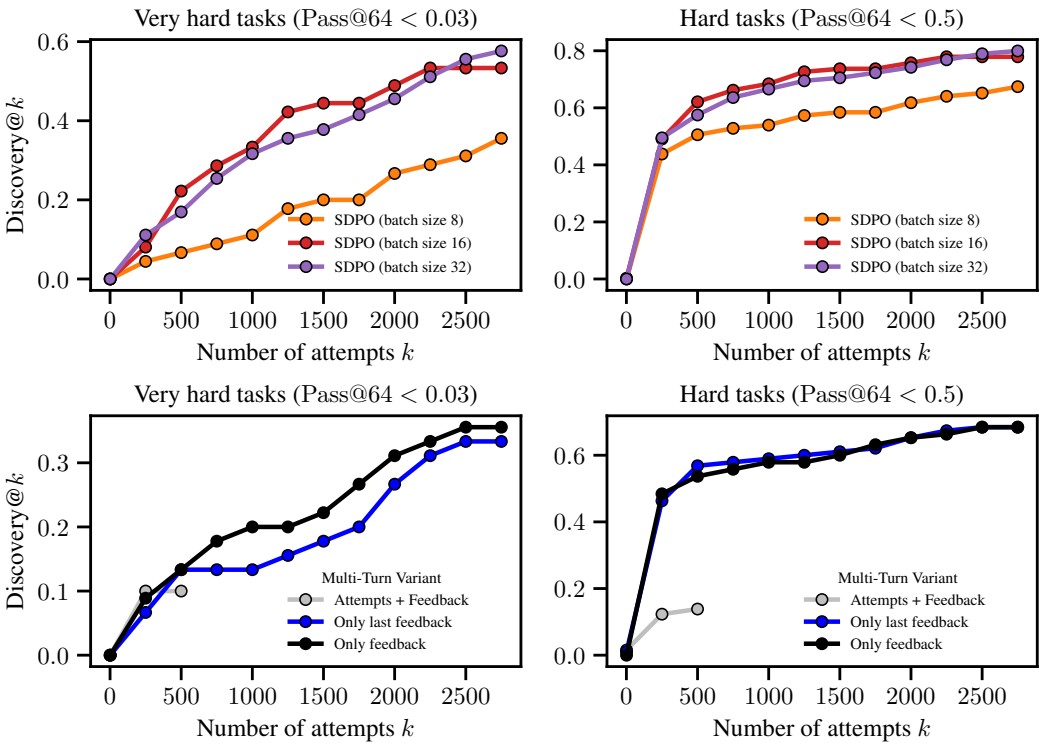

*Figure 19.* **Ablations self-distillation at test-time on very hard (left) and hard (right) tasks. Upper:** Impact of SDPO batch size on pass@$k$ curves. While smaller batch sizes (8 and 16) can lead to slightly earlier discoveries at very low generation budgets ($k < 2^6$), larger batch sizes (16, 32) result in more stable updates that significantly improve the discovery rate as the budget scales. **Lower:** Comparison of multi-turn reprompting templates on very hard (left) and hard (right) tasks. The "Only feedback" template concatenates the feedback from previous attempts using a first-in, first-out sliding window. The "Attempts + Feedback" template concatenates the full turn, also using a sliding window. The "Only last feedback" keeps only the last feedback turn. Including only the feedback substantially outperforms concatenating full conversations. Concatenating feedback across turns significantly improves discovery for very hard questions. As keeping the attempts in-context underperformed already substantially at 500 generations, we did not continue those runs further.

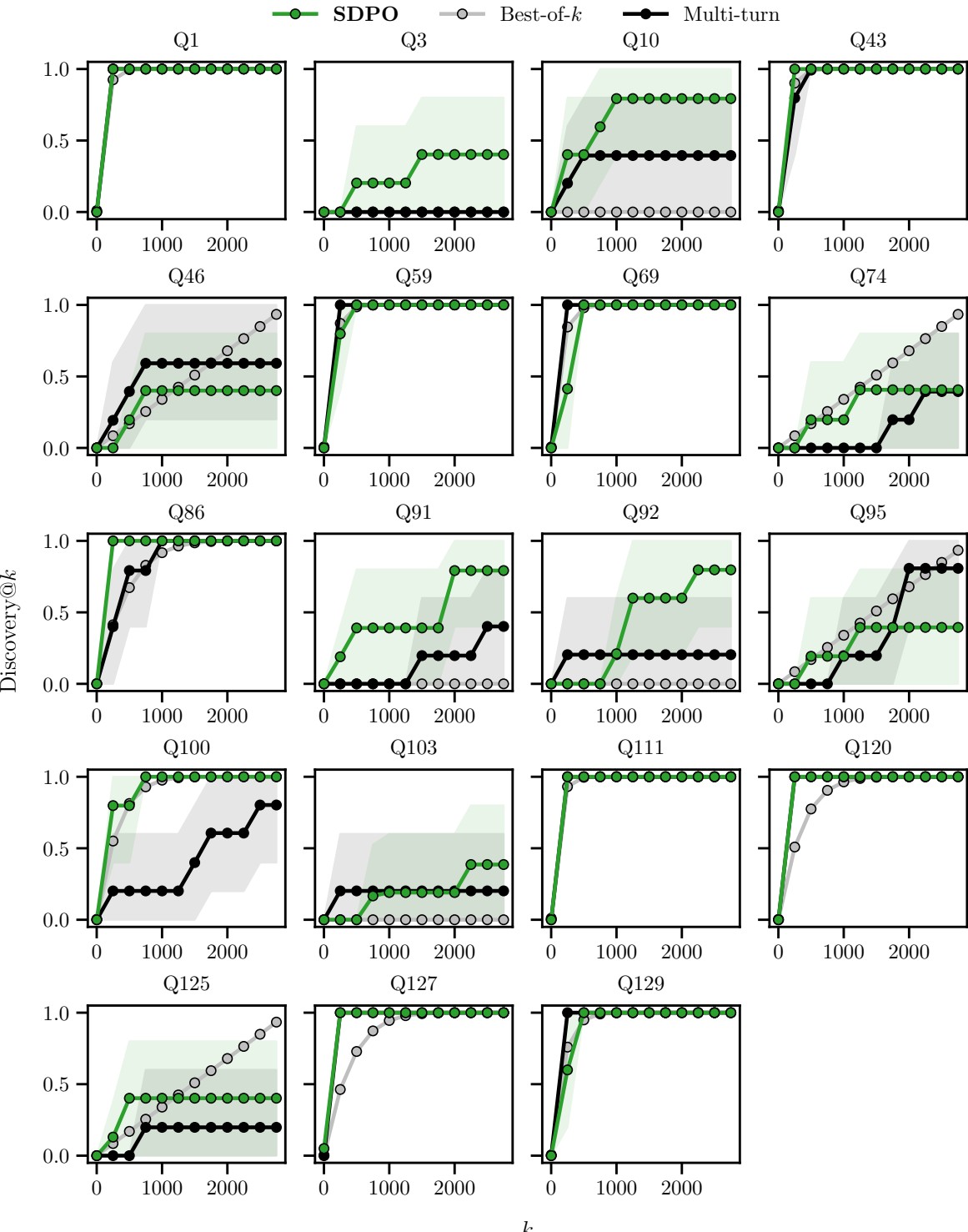

*Figure 20.* **Individual task results self-distillation at test-time.** Discovery@$k$ for each of the 19 questions evaluated in Section 5. In most cases, SDPO finds a successful solution significantly earlier than both the base model and the multi-turn baseline. Notably, for one question (Q3) where the base model and the multi-turn baseline maintain a discovery@$k$ of zero for the entire budget up to 2750 , SDPO discovers a solution after 321 attempts. Curves represent the mean and 90% confidence intervals across 5 random seeds per question.

| Question | Initial Teacher Accuracy (%) |
|---|---|
| 1 | 0.00 |
| 3 | 0.00 |
| 10 | 0.00 |
| 43 | 6.25 |
| 46 | 0.00 |
| 59 | 0.00 |
| 69 | 3.12 |
| 74 | 0.00 |
| 86 | 0.00 |
| 91 | 0.00 |
| 92 | 0.00 |
| 95 | 0.00 |
| 100 | 0.00 |
| 103 | 0.00 |
| 111 | 0.00 |
| 120 | 0.00 |
| 125 | 0.00 |
| 127 | 1.23 |
| 129 | 0.06 |

*Table 11.* **Average accuracy of the retrospective teacher at the first step for each question.** These scores represent the percentage of successful solutions generated when the base model is reprompted with feedback in a single-turn interaction. For the majority of these hard and very hard tasks, the teacher accuracy is near or exactly 0%. Despite this, the self-distilled token-level advantages are sufficiently rich for SDPO to iteratively refine its policy and solve these questions over successive updates.

# I Experiment Details

## I.1 Technical setup

All experiments were conducted on a single node equipped with four NVIDIA GH200 GPUs, for a total of 378GB VRAM. Our environment is built on top of the NVIDIA PyTorch container `nvcr.io/nvidia/pytorch:25.02-py3`, with CUDA 12.8 and PyTorch v2.7.0.

Our implementation is based on the `verl` library (Sheng et al., 2025). We use PyTorch Fully Sharded Data Parallel (FSDP2) for distributed training. For rollout generation, we employ `vLLM` (Kwon et al., 2023), which enables efficient batched inference on the multi-GPU node.

## I.2 Hyperparameters

We summarize hyperparameters used for SDPO in Table 12 and those used for GRPO in Table 13.

| Parameters | Without Feedback Section 3 | With Feedback Section 4 | TTT Section 5 |
|---|---|---|---|
| **General** | | | |
| Model | Qwen/Qwen3-8B allenai/Olmo3-7B-Instruct | Qwen/Qwen3-8B | Qwen/Qwen3-8B |
| Thinking | False | False | False |
| **Data** | | | |
| Max. prompt length | 2048 | 2048 | 2048 |
| Max. response length | 8192 | 8192 | 8192 |
| **Batching** | | | |
| Question batch size | 32 | 32 | 1 |
| Mini batch size | 32 | 1 | 1 |
| Number of rollouts | 8 | 8 | 16 |
| **Rollout** | | | |
| Inference engine | vllm | vllm | vllm |
| Temperature | 1.0 | 1.0 | 1.0 |
| **Validation** | | | |
| Number of rollouts | 16 | 4 | - |
| Temperature | 0.6 | 0.6 | - |
| Top-$p$ | 0.95 | 0.95 | - |
| **SDPO loss** | | | |
| Top-$K$ distillation | 100 | 20 | 20 |
| Distillation divergence | Jensen–Shannon | Reverse-KL | Reverse-KL |
| Clip advantages | – | – | 5.0 |
| Teacher-EMA update rate | 0.05 | 0.01 | 0.01 |
| Rollout importance sampling clip | 2 | 2 | 2 |
| **Training** | | | |
| Optimizer | AdamW | AdamW | AdamW |
| Learning rate | $1 \times 10^{-5}$ (constant) | $1 \times 10^{-6}$ (constant) | $1 \times 10^{-6}$ (constant) |
| Warmup steps | 10 | 0 | 0 |
| Weight decay | 0.01 | 0.01 | 0.01 |
| Gradient Clip Norm | 1.0 | 1.0 | 1.0 |

*Table 12.* Hyperparameters used for **SDPO** for each experimental setup.

I.2.1 DETAILS ON HYPERPARAMETER SELECTION (SECTION 3)

For GRPO in the experiments in Section 3, we perform a grid search over learning rates $\{10^{-5}, 10^{-6}\}$ and minibatch sizes $\{8, 32\}$. For on-policy GRPO, we search over the same learning rates while fixing the minibatch size to 32. For SDPO, we grid-search over KL variants (forward KL, Jensen–Shannon), learning rates $\{10^{-5}, 10^{-6}\}$, and minibatch sizes $\{8, 32\}$. For each method (GRPO, on-policy GRPO, and SDPO), we select a *single* hyperparameter configuration that achieves the highest validation accuracy within the first 5 hours of training, evaluated across all datasets and models used in Section 3. We further report results obtained by selecting the optimal hyperparameter configuration separately for each model and dataset in Table 2.

### I.3 User Templates

For multiple-choice questions and tool use, the model must be prompted in a task-specific manner. We therefore provide the prompt templates used for these settings below.

```
Given a question and four options, please select the right answer. Respond in the
    following format:
<reasoning>
...
</reasoning>
<answer>
...
</answer>

For the answer, only output the letter corresponding to the correct option (A, B, C,
    or D), and nothing else. Do not restate the answer text. For example, if the
    answer is "A", just output:
<answer>
A
</answer>
```

*Listing 1.* **System prompt: Multiple Choice Questions**

```
{question}
Please reason step by step.
```

*Listing 2.* **User prompt: Multiple Choice Questions**

```
You are a helpful function-calling AI assistant. You are provided with function
    signatures within <functions></functions> XML tags. You may call one or more
    functions to assist with the user query. Output any function calls within <
    function_calls></function_calls> XML tags. Do not make assumptions about what
    values to plug into functions.
```

*Listing 3.* **System prompt: Tool use**

```
Your task is to answer the user's question using available tools.
You have access to the following tools:
Name: Axolotl
Description: Collection of axolotl pictures and facts
Documentation:
getRandomAxolotlImage: Retrieve a random axolotl image with information on the image
    source.
Parameters: {}
Output: Successful response.
 - Format: application/json
 - Structure: Object{url, source, description}
```

```
searchAxolotlImages: Search for axolotl images based on specific criteria such as
    color, gender, and size.
Parameters: {"color": "string. One of: [wild, leucistic, albino]. The color of the
    axolotl (e.g., 'wild', 'leucistic', 'albino', etc.).", "gender": "string. One of:
    [male, female]. The gender of the axolotl ('male', 'female').", "size": "string.
    One of: [small, medium, large]. The size of the axolotl ('small', 'medium', 'large
    ').", "page": "integer. The page number for pagination purposes."}
Output: Successful response.
 – Format: application/json
 – Structure: Object{results: Array[Object{url, source, description}], pagination:
    Object{current_page, total_pages, total_results}}
getAxolotlFacts: Retrieve interesting facts about axolotls such as their habits,
    habitats, and physical characteristics.
Parameters: {"category": "string. One of: [habits, habitat, physical characteristics].
     The category of facts to retrieve (e.g., 'habits', 'habitat', 'physical
    characteristics').", "limit": "integer. The maximum number of facts to return."}
Output: Successful response.
 – Format: application/json
 – Structure: Array[Object{fact, source}]

Use the following format:
Thought: you should always think about what to do
Action: the action to take, should be one of the tool names.
Action Input: the input to the action, must be in JSON format. All of the action input
    must be realistic and from the user.

Begin!
Question: Hey, can you show me a random picture of an axolotl?
```

*Listing 4.* **Example user prompt: Tool use**

| Parameters | Experiment 1 |
|---|---|
| | Section 3 |
| **General** | |
| Model | Qwen/Qwen3-8B |
| | allenai/Olmo3-7B-Instruct |
| Thinking | False |
| **Data** | |
| Max. prompt length | 2048 |
| Max. response length | 8192 |
| **Batching** | |
| Question batch size | 32 |
| Mini batch size | 8 (default) / 32 (on-policy) |
| Number of rollouts | 8 |
| **Rollout** | |
| Inference engine | vllm |
| Temperature | 1.0 |
| **Validation** | |
| Temperature | 0.6 |
| Top-$p$ | 0.95 |
| Number of rollouts | 16 |
| **Loss** | |
| $\epsilon$-high | 0.28 |
| Rollout importance sampling clip | 2 |
| KL coefficient ($\lambda$) | 0.0 |
| **Training** | |
| Optimizer | AdamW |
| Learning rate | $1 \times 10^{-6}$ (default) / $1 \times 10^{-5}$ (on-policy) |
| Warmup steps | 10 |
| Weight decay | 0.01 |
| Gradient Clip Norm | 1.0 |

*Table 13.* Hyperparameters used for **GRPO**.

# J   Qualitative Examples

## J.1   Visualization of Advantages

Figure 21 compares the advantages of SDPO and GRPO in a representative example.

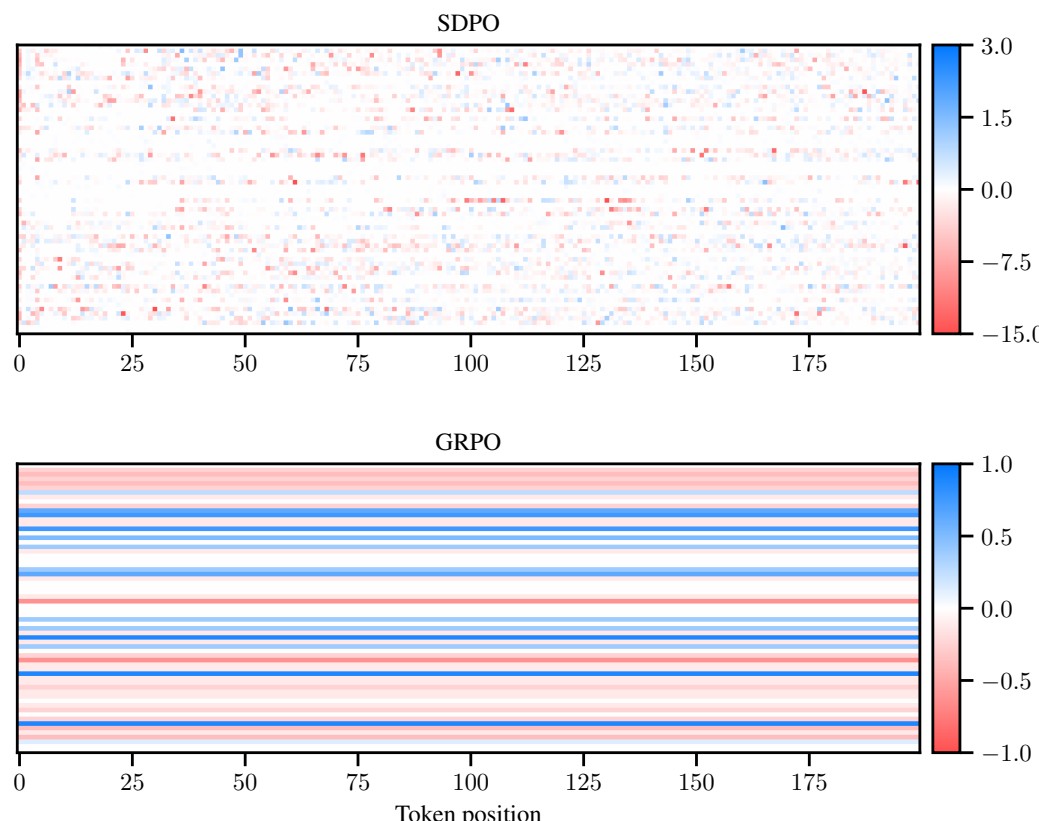

*Figure 21.* Visualization of advantages in SDPO and GRPO with Olmo3-7B-Instruct in a batch from the Chemistry task of Section 3. Each row corresponds to the beginning of a response. The color indicates the advantage value at that token position, with positive advantages shown in blue and negative advantages shown in red.

## J.2   Examples

Below, we show an example from training SDPO on LCBv6 using Qwen3-8B.

```
[Prompt]

You are a coding expert. You will be given a coding problem, and you need to write a correct Python program
    that matches the specification and passes all tests. The time limit is 1 second. You may start by
    outlining your thought process. In the end, please provide the complete code in a code block enclosed with
    ``` ```.

You are given a binary string s of length n, where:

'1' represents an active section.
'0' represents an inactive section.

You can perform at most one trade to maximize the number of active sections in s. In a trade, you:

Convert a contiguous block of '1's that is surrounded by '0's to all '0's.
Afterward, convert a contiguous block of '0's that is surrounded by '1's to all '1's.

Return the maximum number of active sections in s after making the optimal trade.
Note: Treat s as if it is augmented with a '1' at both ends, forming t = '1' + s + '1'. The augmented '1's do
```

```
    not contribute to the final count.

Example 1:

Input: s = "01"
Output: 1
Explanation:
Because there is no block of '1's surrounded by '0's, no valid trade is possible. The maximum number of active
    sections is 1.

Example 2:

Input: s = "0100"
Output: 4
Explanation:

String "0100" -> Augmented to "101001".
Choose "0100", convert "101001" -> "100001" -> "111111".
The final string without augmentation is "1111". The maximum number of active sections is 4.

Example 3:

Input: s = "1000100"
Output: 7
Explanation:

String "1000100" -> Augmented to "110001001".
Choose "000100", convert "110001001" -> "110000001" -> "111111111".
The final string without augmentation is "1111111". The maximum number of active sections is 7.

Example 4:

Input: s = "01010"
Output: 4
Explanation:

String "01010" -> Augmented to "1010101".
Choose "010", convert "1010101" -> "1000101" -> "1111101".
The final string without augmentation is "11110". The maximum number of active sections is 4.

Constraints:

1 <= n == s.length <= 10^5
s[i] is either '0' or '1'

Your solution should have the following signature: ```python
def maxActiveSectionsAfterTrade(s: str) -> int:
```
```

[Response]

<think>

</think>

To solve this problem, we need to understand the structure of the string and the trade operation. Here's the
    plan:

### Key Insights:
1. We are given a binary string `s`, and we are allowed **at most one trade**.
2. A **trade** consists of two steps:
   - Convert a contiguous block of `'1'` that is **surrounded by `'0'`** into `'0'` (i.e., flip it).
   - Then convert a contiguous block of `'0'` that is **surrounded by `'1'`** into `'1'` (i.e., flip it).
3. The actual count of active sections is the number of `'1'` characters **after** the trade. The augmented
   `'1'` at the beginning and end do not count in the final result.

### Strategy:
- We simulate the trade by considering each possible way to flip a block of `'1'` (bounded by `'0'`) and then a
    block of `'0'` (bounded by `'1'`).
- For each possible valid block, simulate the trade and compute the resulting number of `'1'` characters.
- We only need to consider the original string `s`, not the augmented one (since we can simulate the trade
    directly on `s` by checking boundaries).

We'll use a sliding window technique to find contiguous blocks of `'1'` and `'0'` and simulate the trade.

---

### Implementation Steps:
1. Simulate the trade by checking for a block of `'1'` surrounded by `'0'` or at the edges (we simulate the augmented `1` at both sides).
2. For each such block, simulate converting it to `'0'`, then convert a `'0'` block (again, surrounded by `'1'`) to `'1'`.
3. Count the number of `'1'` in the final string and track the maximum.

---

### Python Code:
```python
def maxActiveSectionsAfterTrade(s: str) -> int:
    if not s:
        return 0

    max_ones = 0
    n = len(s)

    # We are simulating the augmented '1's on both ends, so we treat the boundaries as '1's
    # So '0' at index 0 or n-1 is considered surrounded by '1's

    # Step 1: Find all candidate positions for the first part of the trade
    # (convert a block of 1s surrounded by 0s into 0s)
    for i in range(n):
        if s[i] == '1':
            start = i
            while i < n and s[i] == '1':
                i += 1
            end = i - 1
            # Check if this block is surrounded by '0's
            left_boundary = (start == 0 or s[start - 1] == '0')
            right_boundary = (end == n - 1 or s[end + 1] == '0')
            if left_boundary and right_boundary:
                # This is a candidate block of '1's to flip
                # Now, look for a block of '0's surrounded by '1's in the remaining string
                # Replace the '1' block with '0's
                # Now find a '0' block surrounded by '1's
                # We can simulate the remaining string
                temp = list(s)
                for j in range(start, end + 1):
                    temp[j] = '0'

                # Now find a contiguous block of '0's surrounded by '1's
                max_trade_ones = 0
                for k in range(n):
                    if temp[k] == '0':
                        start0 = k
                        while k < n and temp[k] == '0':
                            k += 1
                        end0 = k - 1
                        # Check if this block is surrounded by '1's
                        left_0 = (start0 == 0 or temp[start0 - 1] == '1')
                        right_0 = (end0 == n - 1 or temp[end0 + 1] == '1')
                        if left_0 and right_0:
                            # Flip this block of '0's to '1's
                            for j in range(start0, end0 + 1):
                                temp[j] = '1'
                            # Count the number of '1's
                            count = sum(temp)
                            max_ones = max(max_ones, count)
                            break  # Only take the first valid block for this trade
            # After trying this '1' block, restore the original string
            # No need, we just used a copy
    return max_ones
```

---

### Optimization Note:
The above approach is **not optimal** for very large strings. A more efficient method would involve precomputing the positions of blocks of `'1'` and `'0'`, and checking valid trades between these blocks in linear time.

But with the constraints given (up to 10^5), we need to ensure linear or near-linear performance. The current code is correct but **not efficient** for large inputs.

Would you like me to provide a more optimized version that runs in **O(n)** time and uses only one pass through the string?

```
[Feedback]

b'Runtime Error\nTypeError: unsupported operand type(s) for +: \'int\' and \'str\'\nLine 48 in
    maxActiveSectionsAfterTrade (Solution.py)\n\nLast Executed Input\n"11000"'
```

## J.3  Environment Feedback

We show three examples of feedback in our coding environment, inspired by LeetCode.

```
Test Case 3: Wrong Answer

Input
[[3,6,1],[4,21,4]]

Output
6

Expected
22.87500

Test Case 6: Wrong Answer

Input
[[12,25,3],[3,14,2]]

Output
14

Expected
25.83333
```

*Listing 5.* Example of feedback "Wrong Answer" from our code environment in case of a wrong answer, inspired by LeetCode

```
Runtime Error
MemoryError:
Line 91 in <module> (Solution.py)
Line 25 in solve (Solution.py)

Last Executed Input
10
633 9312
1314 8548
8857 1062
6410 3289
8594 1263
8549 733
3858 5973
... (3 more lines)
```

*Listing 6.* Example of feedback "Memory Error" from our code environment in case of a wrong answer, inspired by LeetCode

```
Runtime Error
IndexError: list index out of range
Line 28 in sortMatrix (Solution.py)

Last Executed Input
[[-1,-1,-1,-1,-1,-1,-1,-1,...
```

*Listing 7.* Example of feedback "Index Error" from our code environment in case of a wrong answer, inspired by LeetCode

## J.4 Illustrative Example

Figure 22 shows an illustrative example of the dense credit assignment in SDPO.

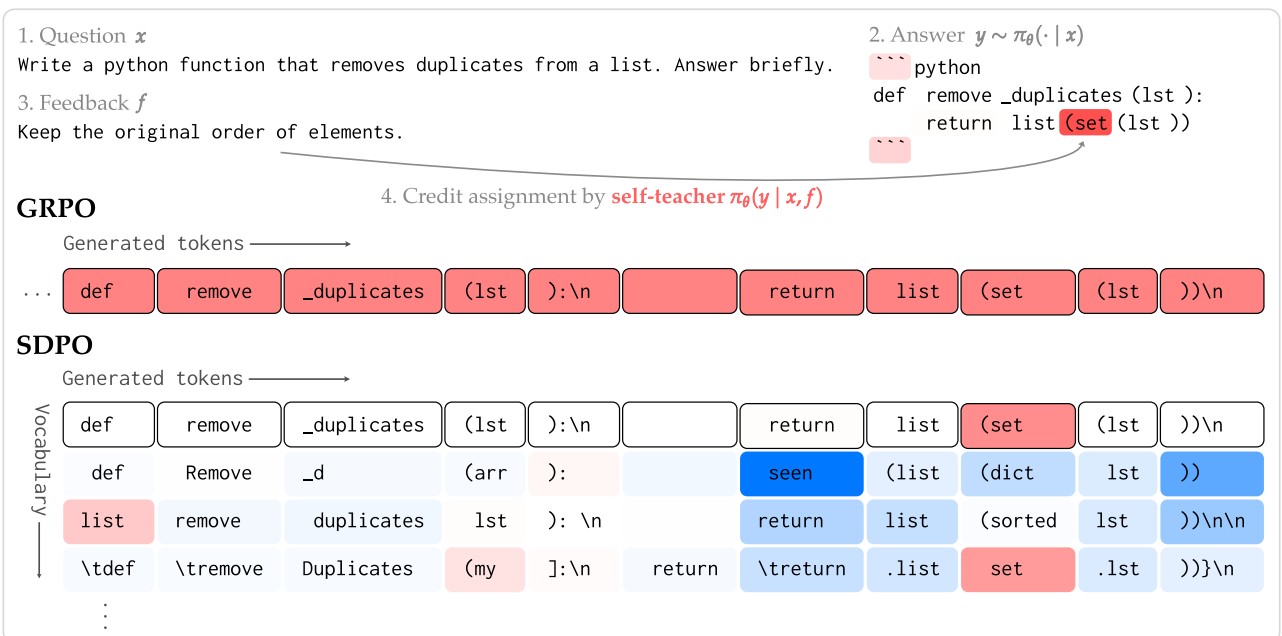

*Figure 22.* **Dense credit assignment through self-teaching in SDPO.** The answer is generated by then model (Qwen3-8B) before seeing the feedback. Then, we re-evaluate the log-probs of the original attempt with the self-teacher after seeing the feedback. We show the per-token $\log\left(\mathbb{P}(\text{self-teacher})/\mathbb{P}(\text{student})\right)$, with red indicating negative values (self-teacher disagrees), blue indicating positive values (teacher reinforces), and white indicating values around zero. Using binary rewards, GRPO would assign the same, negative advantage to all tokens in the sequence. In contrast, SDPO turns the feedback into dense credit assignment across the sequence. The first row shows the tokens of the generated response. The 3 other rows show the top-$k$ logits of the self-teacher that are used during self-distillation, suggesting alternative tokens. Notably, in this example, the self-teacher identifies the error through retrospection without an explicit solution. The credit assignment on the generated sequence, and the alternative top-$k$ logits correctly show that replacing `set` with `dict` maintains the order of elements. Further, in the seventh shown position, the model also identifies an alternative solution path which starts with the `seen` token, instead of directly returning the output. The activation is sparse, identifying where mistakes happen and adjusting to the students' response distribution for specifically these few tokens.

