# OpenReview forum: "Reinforcement Learning via Self-Distillation"
_ICML.cc/2026/Conference — ICML 2026 regular_

### Official Review · Reviewer_wKsQ · 2026-03-05

**Soundness:** 3
**Presentation:** 3
**Significance:** 3
**Originality:** 2
**Overall Recommendation:** 5
**Confidence:** 4

**Summary:**

This paper identifies the credit-assignment bottleneck in reinforcement learning with verifiable rewards (RLVR), where only a scalar outcome reward is available per attempt. The authors formalize a more general setting called Reinforcement Learning with Rich Feedback (RLRF), where environments provide tokenized feedback (e.g., runtime errors, failed unit tests) beyond scalar rewards. They propose Self-Distillation Policy Optimization (SDPO), an on-policy algorithm that uses the current model in two roles: as a "student" that generates attempts and as a "self-teacher" that re-evaluates those attempts conditioned on the received feedback. By minimizing the KL divergence between the student's and the self-teacher's next-token distributions, SDPO obtains dense, logit-level credit assignment without requiring an external teacher or reward model. The authors show SDPO can be derived as a policy gradient algorithm and implemented as a drop-in replacement for GRPO advantages. Experiments span three settings: (1) standard RLVR without rich feedback (science QA, tool use), where successful rollouts serve as implicit feedback for failed ones; (2) RLRF on LiveCodeBench v6 with environment feedback; and (3) test-time self-distillation for accelerating solution discovery on hard problems. SDPO consistently outperforms GRPO in sample efficiency and final accuracy, produces substantially shorter reasoning traces, and achieves 3× faster discovery on hard binary-reward tasks.

**Compliance With Llm Reviewing Policy:**

Affirmed.

**Final Justification:**

The authors have largely addressed my concerns. I appreciate their honesty in the follow-up rebuttal, where they acknowledged that SDPO underperforms GRPO on the LLaMA family of models. This transparency is commendable, though it also reveals a notable limitation: the method appears to rely on the model having a sufficiently strong in-context prior. That said, the additional experiments and clarifications provided in the rebuttal strengthened my confidence in the core contribution, and I have accordingly raised my score from 4 to 5.

**Key Questions For Authors:**

Given that there are several similar works from the same period, how do you view the differences between them and yours? They all seem to involve contextual self-distillation (although an answer isn't required, I'm quite curious).

[1] Self-Distilled Reasoner: On-Policy Self-Distillation for Large Language Models
[2] SELF-DISTILLATION ENABLES CONTINUAL LEARNING
[3] On-Policy Context Distillation for Language Models

**Limitations:**

Yes.

**Strengths And Weaknesses:**

### Strengths
1. The paper provides a clean theoretical derivation showing that the SDPO gradient is a logit-level policy gradient with advantages estimated via the self-teacher.
2. The observation that many verifiable environments already provide rich feedback, and that this feedback is being discarded by scalar-reward methods, is simple but impactful.

## Weaknesses
1. The top-K approximation for logit-level SDPO deserves more scrutiny. The authors fix K=100 throughout, but no ablation over K is provided. It is unclear how sensitive the method is to this choice. Furthermore, choosing the top-K tokens under the student distribution raises a subtle question about on-policy-ness: while SDPO is described as on-policy, the 100th most likely token under the student may carry negligible probability mass, making the advantage signal for such tokens arguably off-policy in spirit.
2. Section 3 uses successful rollouts as feedback for failed attempts. This requires that at least some rollouts in a batch succeed, which means the method still struggles in the same regime where GRPO struggles (all-zero reward batches).
3. The choice of evaluation tasks in Section 3 (line 181–182) is described as "tasks on which the model has not been explicitly fine-tuned," but the rationale is not well-motivated.
4. The paper does not compare against several natural baselines in the rich-feedback setting. For instance, the Thinking Machines Lab blog [1] proposes a related approach earlier where distillation is performed on the actually generated tokens rather than top-1 greedy tokens. The token-level SDPO ablation in Section 4.2 uses only the most likely token at each position, which corresponds to greedy decoding rather than the on-policy generated token.
5. The paper lacks a comparison where a stronger (larger) model is used as teacher in place of the self-teacher. This would help contextualize when self-distillation is preferable to simply using a bigger model, and how much of the gap remains.
6. Given that different model families have different model priors for RL, have you tried on the Llama series?

[1] Lu, Kevin and Thinking Machines Lab, "On-Policy Distillation", Thinking Machines Lab: Connectionism, Oct 2025.

---

> ### Author Rebuttal · Authors · 2026-03-30
>
> > The top-K approximation for logit-level SDPO deserves more scrutiny...
>
> To answer your question we ran an ablation over $K$, extending our analysis of credit assignment from Figure 6 (left). Below are results for LCBv6 with 3 seeds through step 60:
>
> | K | Avg acc | Final acc |
> |---|---:|---:|
> | 1 | 41.6% | 46.2% |
> | 20 | 43.3% | 47.7% |
> | 100 | 43.7% | 46.9% |
> | 200 | 44.0% | 47.8% |
>
> As expected, we see that increasing $K$ yields faster learning and then begins to saturate at larger $K$. We initially selected $K$ to trade off memory overhead against coverage of next-token predictions. In almost all cases, 99% of probability mass is contained within the 100 most likely next tokens.
>
> > Choosing the top-K tokens under the student distribution raises a subtle question about on-policy-ness...
>
> Thank you for highlighting this important point. Since submission, we found a mistake in the original statement of the gradient in Proposition 2.1 and have corrected it in the revision. The implementation used in all experiments already follows the corrected form: the gradient contains an inner expectation over $\hat{y_t} \sim \pi_\theta(\cdot \mid x, y_{1,\dots,t-1})$, rather than an inner sum over $\hat{y_t}$. Together with the tail term in our top-$K$ approximation, this makes the gradient scale independent of $K$ and preserves on-policy weighting under the current policy.
>
> > Section 3 uses successful rollouts as feedback for failed attempts...
>
> In Section 3, we evaluate the standard RLVR setting, showing that SDPO can already improve credit assignment *even without additional environment feedback*. Section 4 then shows that richer environment feedback yields further gains. Section 5 directly studies the regime where solutions have not yet been found, and shows that SDPO also accelerates learning there. We performed an additional joint training run validating this, which we describe in our response to hEpS.
>
> > Choice of evaluation tasks in Section 3
>
> We primarily selected tasks for which the model had not already been extensively tuned, i.e., tasks where the GRPO baseline still shows substantial gains. This provides enough headroom to distinguish methods, rather than evaluating in a near-saturated regime.
>
> > Baselines in the rich-feedback setting & policy-gradient implementation of distillation
>
> Baselines are especially important to us. To our knowledge, we test against the strongest baselines for rich-feedback learning, including in-context learning and SFT on teacher generations.
> There are several ways to implement on-policy self-distillation, analogous to different implementations of on-policy distillation. We view those as ablations of SDPO.
>
> To answer your question, we ablated the implementation of Lu et al. on LCB and did not observe a meaningful difference relative to $K=1$. This is not surprising, since this policy gradient corresponds to a single-sample Monte Carlo estimate of the SDPO gradient.
>
> > Comparison to distillation where a stronger model is used as teacher
>
> SDPO (self-distillation) and distillation are complementary, each addressing a different regime: in self-distillation, no stronger teacher or expert solutions are available in advance. Instead, the policy improves through iterative interaction with the environment. Throughout training, the model never receives supervision from a stronger model, yet the final student substantially exceeds the initial self-teacher (e.g., Figure 6, right). This is qualitatively different from standard distillation, where performance is typically bounded by teacher quality. We do not expect self-distillation to outperform direct distillation when a stronger teacher is already available. Rather, in this case, one might first distill from the stronger teacher and then apply self-distillation to *improve further*. We will clarify this distinction in the revision.
>
> > Other model families
>
> We evaluate SDPO on both Qwen and Olmo model families, which provides meaningful coverage across architectures. We did not include Llama because the strongest Llama checkpoints are older and less competitive on benchmarks such as LCB.
>
> > Given that there are several similar works from the same period, how do you view the differences between them and yours?
>
> Thank you for this question. Although these concurrent works are algorithmically related, the problem setting is different. OPSD and SDFT assume expert solutions are available up front and condition the teacher on those solutions. In contrast, our setting is genuinely RL: answers are not known in advance, and the policy must improve through experience. Compared with OPCD, we provide a substantially broader empirical study across RLVR and rich-feedback settings, include extensive ablations, introduce a novel discovery setting where a model can improve at test time on a single hard question, and make the connection to policy gradient explicit.
>
> We are thankful for your detailed review. We will be happy to answer any further questions.

---

> > ### Author Rebuttal · Reviewer_wKsQ · 2026-04-03
> >
> > > Other model families
> >
> > Since the Llama series exhibits markedly different behaviors compared to the Qwen series, I am curious about the effectiveness of SDPO on the Llama series.

---

> > > ### Author Response · Authors · 2026-04-06
> > >
> > > Thank you again for your review. We appreciate your emphasis on evaluating across a broad range of models. To this end we have expanded our evaluation as follows:
> > >
> > > * We evaluate Llama-3.2-3B-Instruct on the Chemistry task
> > > |Base|GRPO|SDPO|
> > > |---:|---:|---:|
> > > |0.0%|75.1%|62.7%|
> > >
> > >   Although both GRPO and SDPO are improving significantly and stably throughout training ([**training curves**](https://imgur.com/a/fuY2uW9)), GRPO outperforms SDPO by a large margin.
> > >
> > >   We hypothesize that this is due to the relatively weaker in-context learning ability of Llama-3.2-3B compared to our other evaluated models, Qwen3-8B and Olmo3-7B. As illustrated in Figures 5 and 17, SDPO outperforms GRPO with stronger models, while underperforming it with weaker models. We already discuss this prominently in our limitations, but believe that this scaling trend suggests that SDPO may outperform GRPO on future model families, even at small model sizes.
> > >
> > > * To verify that our scaling results not only hold for LiveCodeBench (with rich feedback), but also for Chemistry (without rich feedback), we performed an additional scaling experiment with Qwen3 on Chemistry (5h):
> > > |Model size|Base|GRPO|SDPO|Delta|
> > > |---|---:|---:|---:|---:|
> > > |0.6B|27.7%|71.5%|53.2%|-34.0 pts|
> > > |1.7B|41.8%|73.5%|68.7%|-4.8 pts|
> > > |4B|43.1%|74.7%|77.8%|+3.1 pts|
> > > |8B|41.2%|74.5%|80.9%|+6.4 pts|
> > >
> > >   Alongside our scaling studies with Qwen3 and Qwen2.5 on LCB (Figures 5 and 17), this supports our hypothesis that SDPO becomes more favorable with stronger models.
> > >
> > > * We evaluate two different models (Qwen3-8B and Olmo3-7B) across Section 3. To cover multiple models in our analysis from Section 4 as well, we additionally evaluated Olmo3-7B on LCB (Section 4) and obtained results consistent with those for Qwen3. We ran each method for 10h wall-clock time:
> > > |Model|Base|GRPO|SDPO|
> > > |---|---:|---:|---:|
> > > |Olmo3-7B-Instruct|29.0%|34.2%|43.4%|
> > > |Qwen3-8B (from paper)|27.9%|41.2%|48.8%|
> > >
> > > Overall, we believe that these experiments further expand and support the original claims in our paper.
> > >
> > > Thank you again for your thoughtful follow-up. We hope that our additional results address your remaining concern. If so, we would be very grateful if you would consider updating your score. We cannot update the paper during rebuttal, but we will incorporate all of these additional results into the final camera-ready version.

---

### Official Review · Reviewer_hEpS · 2026-03-12

**Soundness:** 2
**Presentation:** 3
**Significance:** 3
**Originality:** 3
**Overall Recommendation:** 1
**Confidence:** 5

**Summary:**

The paper introduces Self-Distillation Policy Optimization (SDPO), a new post-training algorithm for large language models that addresses a core limitation of existing reinforcement learning approaches. Standard RLVR methods like GRPO only learn from sparse scalar rewards, which creates a severe credit assignment bottleneck, especially when all rollouts in a group receive the same reward. The authors first formalize a richer setting called Reinforcement Learning with Rich Feedback (RLRF), where environments return tokenized feedback (such as runtime errors or judge evaluations) rather than just a scalar. SDPO exploits this by treating the current model, conditioned on the feedback it just received, as a "self-teacher." The self-teacher re-evaluates the original rollout in hindsight and provides dense, token-level credit assignment through a KL distillation loss, with no need for an external teacher model. The algorithm is a minimal modification to standard RLVR pipelines, simply replacing how advantages are computed. SDPO is evaluated on scientific reasoning and tool-use tasks without rich feedback (using successful rollouts as implicit feedback), on competitive programming with LeetCode-style error feedback, and on a test-time training setting where the model must discover solutions to very hard questions. In all settings, SDPO consistently outperforms a strong GRPO baseline in accuracy and sample efficiency while also producing significantly shorter, more concise responses.

**Compliance With Llm Reviewing Policy:**

Affirmed.

**Final Justification:**

Thank you for the additional results and reproduction details on Chemistry domain. The rebuttal improves my confidence in the chemistry experiments, but I remain concerned about practical reproducibility on widely available hardware, since the provided recipe appears tied to a GH200 (or latest gen hardware) setup rather than a more accessible configuration (e.g. A100).

More importantly, my main remaining concern is scope: the paper makes claims suggesting applicability to domains such as math/reasoning, but provides no direct math experiments (in the abstract, the authors mentioned domains like "math and code"). This matters because self-distillation methods do not appear to reliably outperform GRPO on mathematical reasoning. I ran a lot of math training with SDPO, I think the training rewards went up because the model started memorizing the correct demonstrations for the training problems, but the model can't learn to reason through SDPO, and thus the improvement can't transfer to the validation set.

Recent evidence which I came across during the rebuttals [1] also suggests self-distillation (or similar methods) may degrade reasoning performance in that setting. In my view, the current empirical evidence supports a narrower claim than the paper currently makes. I would encourage the authors to either add direct math evidence or explicitly narrow the scope of their conclusions. For these reasons, I remain strongly negative on acceptance in the current form.

[1]: Why Does Self-Distillation (Sometimes) Degrade the Reasoning Capability of LLMs? https://arxiv.org/pdf/2603.24472v1

**Key Questions For Authors:**

Can the authors provide more intuition for why the token-level advantages computed from a teacher that cannot solve the problem are still informative enough to guide learning? Is it possible that the teacher is identifying useful local edits even when its full generated output would still be wrong, and is there an analysis or visualization that directly supports this?

**Limitations:**

yes

**Strengths And Weaknesses:**

Strengths: The core idea of SDPO is genuinely novel and well-motivated. Rather than relying on a stronger external teacher or a trained reward model, the authors observe that a model conditioned on feedback is already a better version of itself, and they build an elegant on-policy distillation algorithm around that insight. This is a conceptually clean contribution that cleanly bridges distillation and RL, and the connection to policy gradient methods (showing SDPO is equivalent to using self-teacher-derived advantages) makes implementation trivially easy as a drop-in swap for existing RLVR pipelines. The empirical evaluation is thorough and honest. The authors test across multiple settings (with and without rich feedback, and at test time), two model families, and several model sizes. The scaling study is particularly compelling since it shows that SDPO's advantage over GRPO grows with model size, which is an intuitive and important finding given that in-context learning ability scales with model size. The analysis of why SDPO produces shorter, more concise responses is a genuinely interesting finding.

Weaknesses: SDPO's quality is fundamentally bounded by the quality of the environment feedback. The ablation in Table 5 is helpful, but the paper does not deeply explore what happens when feedback is noisy, misleading, or incomplete beyond a single sentence in the limitations section. For instance, if a code environment gives an unhelpful error message or a judge LLM produces inconsistent feedback, the self-teacher's advantage signal could be actively harmful. A more systematic investigation of robustness to feedback quality would significantly strengthen the paper's practical claims. The evaluation setup in Section 3 (without rich feedback) uses successful rollouts from the same batch as implicit feedback for failed ones, which essentially mirrors what GRPO does but with a richer signal. However, this only works when the current policy is already solving some questions in each batch. The paper does not thoroughly characterize how SDPO behaves during the early stages of training when the rollout success rate is near zero and no successful example is available in the batch, which is exactly the regime where RLVR methods are known to struggle the most. The comparison between SDPO and multi-turn sampling in Section 5 is somewhat unbalanced. The multi-turn baseline also receives the same rich feedback in-context, but the model's weights are never updated. The authors frame SDPO's advantage as coming from "compressing context into weights," but it's not fully clear whether the gains come from the weight update itself, from the particular way feedback is incorporated into the reprompting template, or from some other factor.

---

> ### Author Rebuttal · Authors · 2026-03-30
>
> Thank you for your constructive review, and for recognizing our “genuinely novel and well-motivated” contribution and thorough empirical evaluation.
>
> > SDPO's quality is fundamentally bounded by the quality of the environment feedback. The ablation in Table 5 is helpful, but the paper does not deeply explore what happens when feedback is noisy, misleading, or incomplete.
>
> Thank you for highlighting this limitation. To further analyze the robustness of SDPO to imperfect feedback, we performed additional experiments.
>
> First, in the RLVR setting of Section 3, we study noisy rewards by flipping the binary reward assigned to each rollout with probability $p$ (chemistry, Qwen3-8B). We report the highest validation accuracy reached during 5 hours of training, excluding the initial accuracy:
>
> | p | GRPO | SDPO |
> |---|---:|---:|
> | 0.0 | 74.7% | 80.2% |
> | 0.2 | 57.3% | 65.9% |
> | 0.4 | 45.0% | 58.1% |
> | 0.6 | 21.8% | 41.4% |
> | 0.8 | 10.3% | 34.8% |
>
> SDPO is consistently more robust than GRPO, and still improves the policy at $p=0.6$.
>
> We also studied noisy textual feedback on LCB by (1) dropping feedback, (2) shuffling feedback across rollouts, and (3) truncating feedback to the first line:
>
> | Method | Acc |
> |---|---:|
> | GRPO | 41.2% |
> | no noise | 47.6% |
> | truncation | 48.5% |
> | shuffle(0.5) | 47.3% |
> | shuffle(0.75) | 47.3% |
> | shuffle(1.0) | 47.0% |
> | drop(0.5) | 47.2% |
> | drop(0.75) | 47.0% |
> | drop(1.0) | 45.0% |
>
> These results use Qwen3-8B trained to step 60 and averaged across 3 seeds. We observe only modest degradation under truncation, shuffling, and partial dropping, and all variants outperform drop(1.0), where textual feedback is entirely removed. We will include these results in the revision to strengthen the empirical support for SDPO’s robustness.
>
> > The evaluation setup in Section 3 (without rich feedback) uses successful rollouts...
>
> Our goal in Section 3 was to evaluate SDPO in the standard RLVR setting, showing that it can improve credit assignment *even without additional environment feedback*. Section 4 then shows that richer environment feedback yields further gains, and Section 5 studies the harder regime where solutions have not yet been found.
>
> To address your concern more directly, we ran joint training across 3 seeds as in Section 4, but restricted to questions with pass@64 < 0.5, yielding a similarly low-success regime while still training across multiple questions:
>
> | Method | Final acc| Avg acc|
> |---|---:|---:|
> | GRPO | 8.7% | 2.5% |
> | SDPO | 21.7% | 11.7% |
>
> We report validation accuracy on the same subset of questions as in Figure 7 (right). In this regime, GRPO makes only small updates: 70% of gradients up to step 20 have norm 0. By contrast, SDPO maintains a stable gradient norm around 0.5. This indicates that SDPO with rich feedback learns substantially faster than GRPO even when rollout success is near zero.
>
> > The comparison between SDPO and multi-turn sampling in Section 5 is somewhat unbalanced…
>
> Ensuring a fair comparison between SDPO and multi-turn is very important to us. In our experiments, multi-turn uses the same reprompt template as SDPO. The key difference is that SDPO compresses earlier feedback into the weights before proceeding (Figure 11), whereas multi-turn keeps that feedback in context.
>
> We also compared against an alternative multi-turn variant that retains prior attempts in context, but it reaches the context limit much faster and performs worse (Figure 19, right).
> To address your concern, we additionally evaluated a baseline that generates from the self-teacher using only the latest feedback in context. This baseline underperforms SDPO throughout. On very hard tasks, it also underperforms multi-turn, while on hard tasks it only slightly outperforms multi-turn for $k \leq 1500$.
>
> > Can the authors provide more intuition for why the token-level advantages computed from a teacher that cannot solve the problem are still informative enough to guide learning? Is it possible that the teacher is identifying useful local edits even when its full generated output would still be wrong, and is there an analysis or visualization that directly supports this?
>
> Yes, our qualitative findings are consistent with exactly this intuition. Figure 9 shows a representative example with Qwen3-8B: the model initially makes an incorrect assumption about the environment, and after receiving feedback, corrects that local mistake. We will move this visualization into the main paper in the revision.
>
> This "useful local edit" interpretation is also supported by Table 5. Only about 10% of self-teacher generations receive the same feedback as the original student, while more than 55% of self-teacher responses still fail. This suggests that the self-teacher often changes behavior directionally without immediately solving the problem.
>
> ---
>
> We sincerely appreciate your review, and hope that each of your comments was addressed. We are happy to further discuss any potential follow-up.

---

> > ### Author Rebuttal · Reviewer_hEpS · 2026-04-03
> >
> > Thank you for the response, I appreciate these additional results.
> > However, after some attempts and a lot of efforts, I cannot seem to reproduce results from the paper, especially I cannot get SDPO to perform reliably better than vanilla GRPO with the scripts from the supplementary materials, thus I am reducing my rating to 1.
> > I strongly encourage the authors to release the full training recipe, including the Docker container, training data and full scripts, wandb training logs, full trained model weights, and evaluation scripts, to better help the community understand the contribution of this work.

---

> > > ### Author Response · Authors · 2026-04-04
> > >
> > > Thank you for taking the time to reproduce our results. We take reproducibility extremely seriously. To this end, we provide a detailed recipe for reproducing our results. We provide the Dockerfile of our environment, exact scripts for reproducing experiments, an image of W&B logs, and an anonymized checkpoint from the Qwen/Chemistry run. We will share all of these publicly as well.
> > >
> > > **Chemistry reproduction recipe:** All runs were performed on a **single node with 4 GH200 GPUs**. Our updated anonymized code is here: https://anonymous.4open.science/r/SDPO-4516
> > >
> > > For Chemistry, the steps are:
> > > * Build the environment via `env/build_env.slurm`
> > > * Preprocess the data via `python data/preprocess.py --data_source /XXX/datasets/sciknoweval/chemistry`
> > > * Run the experiment via `sh experiments/generalization/run_sdpo_all.sh`
> > > * The HF checkpoint (below) can be validated with `sh experiments/generalization/eval_chemistry_checkpoint.sh`
> > >
> > > Note that the scripts under `experiments/` set additional hyperparameters. Reproducing the result therefore requires using the experiment script directly, rather than invoking lower-level training entrypoints with default arguments. Since the original submission, we have further tuned hyperparameters for both SDPO and GRPO. The linked anonymized code therefore reproduces our latest results, which are higher than those in the original submission, but don’t change the qualitative conclusion.
> > >
> > > Other experiments can be reproduced analogously from the same codebase (more details in the Readme).
> > >
> > > To further support reproducibility, we are also sharing:
> > > * [**📊 W&B screenshot for the Chemistry run ↗️**](https://imgur.com/a/6uNLHn3) (green: SDPO, gray: GRPO)
> > > * [**📊 Extension of Figure 3 to three seeds ↗️**](https://imgur.com/a/8mmJSBr)
> > > * An anonymous checkpoint for the Chemistry SDPO run (~80.5% test accuracy): https://huggingface.co/anon56948576/qwen3-8b-sdpo-chemistry
> > >
> > > It would have been very helpful to learn about this issue already in your initial review, since that would have allowed us to clarify the exact recipe earlier in the discussion.
> > > We respectfully ask that the paper be evaluated based on the method and empirical evidence in the submission alongside the additional results we provided in response to your review.

---

### Official Review · Reviewer_9LHn · 2026-03-12

**Soundness:** 3
**Presentation:** 3
**Significance:** 3
**Originality:** 3
**Overall Recommendation:** 4
**Confidence:** 3

**Summary:**

This paper studies reinforcement learning for language models in settings where the environment provides not only a scalar reward but also richer tokenized feedback, such as runtime errors and judge comments. The paper formalizes this setting as Reinforcement Learning with Rich Feedback and proposes Self-Distillation Policy Optimization (SDPO), where the current model, when conditioned on feedback, serves as a self-teacher whose next-token distribution is distilled back into the student policy. Empirical results show strong performance of SDPO on several challenging tasks.

**Compliance With Llm Reviewing Policy:**

Affirmed.

**Final Justification:**

The rebuttal addressed my concerns so I maintain my positive score.

**Key Questions For Authors:**

See weakness

**Limitations:**

Yes

**Strengths And Weaknesses:**

** Strength: **

The paper addresses an important problem: the credit-assignment bottleneck of scalar-reward RL for LLMs. The central idea is appealing and reasonably novel. Instead of relying on an external teacher or a learned reward model, the method uses the same policy in hindsight, conditioned on feedback, to produce dense token-level supervision. This makes the method conceptually simple, practically relevant, and easy to integrate into existing RLVR pipelines. The paper evaluates both rich-feedback environments and standard RLVR settings, includes scaling results, analyzes dense and sequence-level credit assignment, studies forgetting, and reports modest compute overhead.



** Weakness: **

(1) Some of the strongest claims are empirical and could use tighter isolation, In the “no environment feedback” setting, SDPO depends on successful within-batch solutions as pseudo-feedback, so it would be useful to quantify how often this signal is actually available, especially early in training when groups may contain no successful attempt.

(2) In the test-time evaluation protocol, the paper filters to hard questions for which at least one of the compared methods finds a solution within 512 steps across 5 seeds. That makes the setting understandable for analysis, but it also conditions on solvability and may overstate the practical strength of the method relative to an unconditional evaluation on the full hard subset.

(3) The baseline is only limited to GRPO.

---

> ### Author Rebuttal · Authors · 2026-03-29
>
> We thank the reviewer for their review, and for noting that our paper addresses an important problem with a conceptually clean and practically relevant method.
>
> > The central idea is appealing and reasonably novel.
>
> We appreciate this assessment. We would like to clarify what we view as the main conceptual and empirical contributions of our work. To our knowledge, we introduce the **first on-policy algorithm for learning from rich feedback** (token sequences instead of scalar rewards). We evaluate this algorithm across diverse settings (scalar rewards, rich feedback, and discovery from rich feedback before a question has ever been solved). We **show that our algorithm outperforms strong baselines** across these settings. Section 5 further demonstrates a **novel application of RL to discovery**, where previous RL algorithms for LLMs do not improve over simple best-of-k sampling. Conceptually, we show that our algorithm can be seen as a natural extension of policy gradient algorithms while bridging the literature of on-policy RL and on-policy distillation.
>
> > Some of the strongest claims are empirical and could use tighter isolation. In the “no environment feedback” setting, SDPO depends on successful within-batch solutions as pseudo-feedback, so it would be useful to quantify how often this signal is actually available, especially early in training when groups may contain no successful attempt.
>
> To answer this question, we computed the fraction of rollout groups with any successes across settings. Note that successes in rollout groups are used by both SDPO and GRPO. In Section 3 (chemistry), initially 70%–80% of groups per batch contain at least one success. In Section 4 (LCB), around 40% of groups contain successes. In Section 5 (discovery in LCB), runs are stopped at the first success; hence prior successes are 0% by construction. We believe that this covers the relevant settings.
>
> Beyond this concrete question, we note that the paper already includes extensive ablations to isolate the components of our framework, including ablations over different types of feedback (Table 5), the specificity of credit assignment (Figure 6, left), and the bootstrapping of the student beyond the initial teacher (Figure 6, right).
>
> In our response to reviewer hEpS, we additionally ablate noisy feedback and evaluate joint training on LCB with only hard questions.
>
> > In the test-time evaluation protocol, the paper filters to hard questions for which at least one of the compared methods finds a solution within 512 steps across 5 seeds. That makes the setting understandable for analysis, but it also conditions on solvability and may overstate the practical strength of the method relative to an unconditional evaluation on the full hard subset.
>
> Thank you for noting this. Concretely, there are 47 questions for which no method discovered a solution by $k=512$ across 5 seeds. We would like to emphasize that this still enables a fair relative comparison between methods. We excluded these questions from evaluation because running all methods until $k=2750$ would have more than tripled our compute cost for limited additional signal. Our original goal of this evaluation was to study (conditional) discovery efficiency. That said, we agree that the current presentation misrepresents the difficulty of the full LCB benchmark. In the revision, we will rescale the y-axis to account for these additional 47 unsolved questions. The relative comparison between methods remains the same, while the absolute scale will better reflect the difficulty of LCB questions.
>
> > The baseline is only limited to GRPO.
>
> This is not the case, though we understand where this impression may come from. Our main baseline in the paper is an improved version of GRPO that incorporates several recent modifications, including asymmetric clipping [1], avoiding biased normalization [2], and rollout off-policy correction [3] (see line 195). Additionally, Appendix H.2.3 reports several further algorithms for LCBv6: GSPO, CISPO, and excluding low-entropy tokens [4]. Based on your feedback, we also ran the GRPO baseline without improvements [1,2,3] for LCBv6, which led to slightly reduced final accuracy: 40.6% vs. 41.2%. In summary, SDPO significantly outperforms strong baselines across all settings in Sections 3, 4, and 5. We will clarify the discussion of the main GRPO baseline in the revised version.
>
> ---
>
> Thank you for taking the time to review our submission and for providing actionable feedback. We hope we were able to address each point by providing new empirical evidence and clarifying existing baselines.
>
> ---
>
> [1] Yu et al. DAPO: An Open-Source LLM Reinforcement Learning System at Scale. In NeurIPS, 2025.
>
> [2] Liu et al. Understanding R1-Zero-Like Training: A Critical Perspective. In COLM, 2025.
>
> [3] Olmo 3. 2025.
>
> [4] Wang et al. Beyond the 80/20 Rule: High-Entropy Minority Tokens Drive Effective Reinforcement Learning for LLM Reasoning. In NeurIPS, 2025.

---

> > ### Author Rebuttal · Reviewer_9LHn · 2026-04-05
> >
> > Thanks for your response. My concern is resolved and I will maintain my positive score.

---

> > > ### Author Response · Authors · 2026-04-06
> > >
> > > Thank you for your follow-up and for your thoughtful review. Since our rebuttal and new experiments appear to have addressed your concerns, we would be very grateful if you would consider updating your score to reflect your current assessment.
> > >
> > > We remain excited about our work’s methodological advance: self-distillation enables policies to learn from rich feedback and leverages that feedback for dense credit assignment. In addition to introducing self-distillation and connecting it to policy gradient methods and on-policy distillation, our paper presents a systematic, large-scale empirical study across several domains, showing when self-distillation is beneficial.

---

### Official Review · Reviewer_H5SN · 2026-03-13

**Soundness:** 4
**Presentation:** 3
**Significance:** 4
**Originality:** 4
**Overall Recommendation:** 5
**Confidence:** 2

**Summary:**

The paper proposes a new algorithm to improve the performance of Reinforcement Learning for post-training of LLMs. The main idea is rather than using a scalar value as the reward, the authors propose making use of the additional feedback available in environments that provide more feedback than just right/wrong. They cleverly provide the question along with the feedback to the same model again as a new query and then look at its generated tokens now given the feedback then apply back propagation using the new values of the originally produced tokens. Their approach achieves impressive results beating GRPO in accuracy and number of iterations.

**Compliance With Llm Reviewing Policy:**

Affirmed.

**Final Justification:**

I believe the paper is strong and a good addition to the field. The authors did a good job addressing my rebuttal and I'm happy to keep my already positive rating.

**Key Questions For Authors:**

- Can you redraw Figure 7 with X-axis to be the wall-clock time of FLOPs instead of number of attempts? I think that’s a more representative metric since the cost of each attempt is not the same in both algorithms.
- Assuming an environment where the feedback might not always be reliable. Do you have any suggestions on how to minimize the effect of that on your method?

**Limitations:**

yes

**Strengths And Weaknesses:**

Strengths:

- The contribution is novel and interesting - tackles an important area as well.
- Drop-in replacement in standard RLVR pipelines without rich feedback still achieves better results
- Impressive results overall.

Weaknesses:

- Results assume high quality feedback (Authors already acknowledged the limitation).
- Lots of forward references to the Appendix in the main text making it hard to grasp the paper without reading the appendix.

---

> ### Author Rebuttal · Authors · 2026-03-30
>
> We thank the reviewer for their positive assessment and are glad they found the paper novel, important, and empirically strong.
>
> > Results assume high quality feedback (Authors already acknowledged the limitation).
>
> We thank the reviewer for highlighting this limitation. To analyze the robustness of SDPO to noisy feedback, we performed additional experiments as part of our rebuttal.
>
> First, we analyze robustness to noisy rewards in the RLVR setting of Section 3. With probability $p$, we flip the binary reward assigned to a rollout, and sweep $p$ (chemistry, Qwen3-8B). We report the highest validation accuracy during 5h of training (excluding initial accuracy):
>
> | p | GRPO | SDPO |
> |---|---:|---:|
> | 0.0 | 74.7% | 80.2% |
> | 0.2 | 57.3% | 65.9% |
> | 0.4 | 45.0% | 58.1% |
> | 0.6 | 21.8% | 41.4% |
> | 0.8 | 10.3% | 34.8% |
>
> SDPO is consistently more robust than GRPO, and still improves the policy at $p=0.6$.
>
> We also studied noisy textual feedback on LCB by (1) dropping feedback, (2) shuffling feedback across rollouts, and (3) truncating feedback to the first line:
>
> | Method | Acc |
> |---|---:|
> | GRPO | 41.2% |
> | no noise | 47.6% |
> | truncation | 48.5% |
> | shuffle(0.5) | 47.3% |
> | shuffle(0.75) | 47.3% |
> | shuffle(1.0) | 47.0% |
> | drop(0.5) | 47.2% |
> | drop(0.75) | 47.0% |
> | drop(1.0) | 45.0% |
>
> These results are with Qwen3-8B until step 60, averaged across 3 seeds. We do not observe severe degradation under any noise type, and all variants outperform drop(1.0), where textual feedback is entirely removed. We will add these robustness results to the revision.
>
> More broadly, Sections 3-5 show that SDPO is effective with binary rewards alone, binary rewards plus rich feedback, and rich feedback alone, suggesting that it does not depend on perfectly informative feedback.
>
> > Lots of forward references to the Appendix in the main text making it hard to grasp the paper without reading the appendix.
>
> We thank the reviewer for this important feedback. We agree that the paper can be made more self-contained. In the revision, we will move the most important appendix material into the main text, including figures/tables that build intuition for SDPO and clarify its relation to prior work. This will substantially reduce forward references.
>
> > Can you redraw Figure 7 with X-axis to be the wall-clock time of FLOPs instead of number of attempts?
>
> We agree that both sample-efficiency and compute-efficiency are useful ways to assess discovery efficiency, i.e., how effectively a method learns from feedback.
> In Figure 7, we chose to emphasize sample-efficiency because, in our code environment, evaluation is roughly $3\times$ more expensive (wall-clock) than generation. Moreover, sample-efficiency is less tied to implementation-specific assumptions than FLOPs or wall-clock time.
>
> That said, SDPO’s extra cost is modest: relative to best-of-k, its per-attempt FLOPs (excluding environment evaluation) are approximately
> $2.3\times$ larger under reasonable simplifying assumptions (see below), whereas multi-turn methods are much more expensive due to long contexts. Combined with the roughly $3\times$ sample-efficiency gain in Figure 7, this suggests SDPO can remain favorable under compute-based metrics as well.
>
> > Assuming an environment where the feedback might not always be reliable. Do you have any suggestions on how to minimize the effect of that on your method?
>
> Our new ablations suggest SDPO is already fairly robust to noisy feedback. We suspect this is because SDPO does not directly optimize against raw feedback; instead, the model first conditions on the feedback in context and only then converts it into a learning signal. In practice, robustness can likely be improved further by preprocessing feedback, filtering clearly irrelevant or low-quality feedback, and combining SDPO with GRPO when reliable scalar rewards are available.
>
> ---
>
> We sincerely appreciate your review, and hope that each of your comments was addressed. We are happy to further discuss any potential follow-up.
>
> ---
>
> Using the common assumption that a full forward pass with $N$ parameters and $T$ tokens costs $\approx 2NT$ FLOPs, the per-attempt FLOPs are:
> * Best-of-k: $2N(T_{prompt}+T_{out})$
> * Multi-turn: $2N(T_{context})$
> * SDPO: student sample + self-teacher forward + student backward
>   $= 2N(T_{prompt}+T_{out}) + 2N(T_{prompt}+T_{out}+T_{feedback}) + 4N(T_{prompt}+T_{out}+T_{feedback})/B$, with batch size $B$.
>
> Using prompt length 500, output length 1k, feedback length 200, context length 40k, and $B=16$:
> * Best-of-k: $2N \times 1500$
> * Multi-turn: $2N \times 40000$ ($\approx 26\times$ best-of-k)
> * SDPO: $2N \times 3412.5$ ($\approx 2.3\times$ best-of-k)

---

> > ### Author Rebuttal · Reviewer_H5SN · 2026-04-03
> >
> > Thank you for answering my questions and for the great work. I'm maintaining my already positive score.
> > I need to mention that I did not make any attempt to reproduce the results and my score is based on your reported results. Just saying that after reading the rest of the reviews, especially the comment from Reviewer hEpS.

---

> > > ### Author Response · Authors · 2026-04-04
> > >
> > > Thank you for your detailed review and thoughtful comments. In the camera-ready version, we will use the additional page to include further results on robustness to feedback noise, discuss the version of Figure 7 with FLOPs on the x-axis, and reduce forward references to the appendix by moving key figures into the main text. We sincerely appreciate your feedback and believe these changes will improve the paper.
> > >
> > > **Regarding reviewer hEpS’s comment on reproducibility**, our response provides detailed step-by-step instructions and supporting artifacts that validate our results. We take reproducibility extremely seriously and plan to release these materials publicly. In addition, we are aware of several independent research groups that have reproduced our empirical findings. We are therefore confident in the accuracy of our results, and we hope our detailed reply to reviewer hEpS addresses this concern.

---

### Decision · Program_Chairs · 2026-04-30

**Decision:**

Accept (regular)

**Comment:**

This submission studies the problem of LLM RL training from a scalar reward. This work proposes the idea of self distillation. It uses feedback from the environment to build a denser learning signal through self-distillation. The reviewers find that this is an important problem and that the method is simple and practical. The experiments are broad and show clear gains in several settings. Overall, the rebuttal added extra results on noisy feedback, more backbone models, and implementation details.

The strengths identified by the reviewers:
- 1.	It studies an important problem and presents a clear idea. Several reviewers find that the work addresses a credit-assignment bottleneck in RLVR and proposes an interesting solution (H5SN, 9LHn, wKsQ).
- 2.	It demonstrates the proposed mehtod with strong empirical results. It shows empirical gains in several settings, including rich feedback, standard RLVR, and test-time discovery. All reviewers found the evaluation strong overall (H5SN, 9LHn, hEpS, wKsQ).

The weaknesses
- 1. Reviewers raise concerns on limits across model families and domains. One reviewer remained concerned that the method may work better for some models than others, and that claims should be stated more carefully. However, it is difficult to demonstrate results across multiple families of llms given the compute constraints for most people/ teams in the world.
- 2.	One reviewer heps also raises reproducibility concerns. S/he reports difficulty reproducing results and remained negative, although the authors provided extra details and checkpoints during rebuttal.

Overall, three reviewers give positive ratings (two accepts and one weak ac), and viewed the work as novel and useful. The proposed method addresses an important problem with a simple idea and strong empirical evidence. The rebuttal resolved concerns by adding results on noisy feedback, more baselines, more model families, and more reproducibility details. While one reviewer remained strongly negative, this view is outweighed by the broader positive consensus and the submission's clear strengths. Therefore this work in its current form is recommended with a positive decision.